# A map of the rubisco biochemical landscape

Noam Prywes[1,2], Naiya R. Phillips[3], Luke M. Oltrogge[2,3], Sebastian Lindner[4], Leah J. Taylor-Kearney[5], Yi-Chin Candace Tsai[6], Benoit de Pins[7], Aidan E. Cowan[3,8], Hana A. Chang[5], Renée Z. Wang[5], Laina N. Hall[9], Daniel Bellieny-Rabelo[1,10], Hunter M. Nisonoff[11], Rachel F. Weissman[3], Avi I. Flamholz[12], David Ding[1,2], Abhishek Y. Bhatt[3,13], Oliver Mueller-Cajar[6], Patrick M. Shih[1,5,14,15], Ron Milo[16] & David F. Savage[1,2,3]✉

Rubisco is the primary $CO_2$-fixing enzyme of the biosphere[1], yet it has slow kinetics[2]. The roles of evolution and chemical mechanism in constraining its biochemical function remain debated[3,4]. Engineering efforts aimed at adjusting the biochemical parameters of rubisco have largely failed[5], although recent results indicate that the functional potential of rubisco has a wider scope than previously known[6]. Here we developed a massively parallel assay, using an engineered *Escherichia coli*[7] in which enzyme activity is coupled to growth, to systematically map the sequence–function landscape of rubisco. Composite assay of more than 99% of single-amino acid mutants versus $CO_2$ concentration enabled inference of enzyme velocity and apparent $CO_2$ affinity parameters for thousands of substitutions. This approach identified many highly conserved positions that tolerate mutation and rare mutations that improve $CO_2$ affinity. These data indicate that non-trivial biochemical changes are readily accessible and that the functional distance between rubiscos from diverse organisms can be traversed, laying the groundwork for further enzyme engineering efforts.

Plants, algae and photosynthetic bacteria together fix around 100 gigatons of carbon annually using ribulose-1,5-bisphosphate carboxylase/oxygenase (rubisco)—the most abundant enzyme on Earth[8]. Rubisco catalysis, which is slow compared with many other central carbon metabolic enzymes[2], is thought to limit photosynthesis under common conditions[9]. Rubisco is also prone to a side reaction with oxygen, leading to the hypothesis that this apparent inefficiency is in fact a careful balance of several biochemical trade-offs between rate, affinity and promiscuity[10–13].

Efforts to engineer improvements to rubisco have been hampered by the low throughput of obtaining accurate measurements for its parameters, including catalytic rate for carboxylation ($k_{cat,C}$, called $k_{cat}$ here), $CO_2$ affinity ($K_C$) and specificity for $CO_2$ versus $O_2$ ($S_{C/O}$). A concentrated effort across several decades has produced several hundred biochemical measurements of natural and mutant rubiscos[10–13]. Collection of these measurements has been biased towards vascular plant rubiscos, and the diversity of natural rubiscos remains undersampled. Library screens and rational mutations have been used in the past to increase rubisco activity. These efforts often resulted in improved expression[5] but occasionally led to fundamental biochemical improvements[14,15].

Protein engineering has benefited in recent years from the introduction of machine learning approaches. One goal of such efforts is to train models with labelled protein sequence–function data from high-throughput functional screens[16–21]. Enzyme engineering with machine learning presents a further challenge: ideally, functional data would be decomposed into individual catalytic parameters measured in high throughput either in vitro[22] or in vivo[20].

Here we have developed a selection assay in *Escherichia coli* to estimate the carboxylation fitness of more than 99% (8,760 of 8,835) of the single-amino acid mutants of the model Form II rubisco from *Rhodospirillum rubrum* (Extended Data Fig. 1). Ribose phosphate isomerase was knocked out to generate *Δrpi*—a strain that grows on glycerol only when it expresses functional rubisco (Extended Data Fig. 2a). We then generated a barcoded library of single-amino acid mutations of the *R. rubrum* rubisco, which we assayed in high throughput using *Δrpi*. By varying the $CO_2$ concentrations of the growth environment, we were able to estimate the effective $CO_2$ affinities of 65% (5,687) of the rubisco variants—a subset of which we went on to validate in vitro. This screen showed a very small minority of mutations that improved affinity for $CO_2$ around threefold. These affinities have never before been observed among bacterial rubiscos, are more typical of the Form I rubiscos found in plants and algae, and indicate that non-trivial alterations to biochemical function are rare, yet readily accessible through mutation.

## Characterization of rubisco variants

The rubisco-dependent *E. coli* strain, *Δrpi*, cannot grow when glycerol is provided as the only carbon source because ribulose-5-phosphate accumulates with no outlet[7]. The combined actions of phosphoribulokinase (PRK, which produces the five-carbon rubisco substrate) and

[1]Innovative Genomics Institute, University of California Berkeley, Berkeley, CA, USA. [2]Howard Hughes Medical Institute, University of California Berkeley, Berkeley, CA, USA. [3]Department of Molecular and Cell Biology, University of California Berkeley, Berkeley, CA, USA. [4]University of Heidelberg, Heidelberg, Germany. [5]Department of Plant and Microbial Biology, University of California Berkeley, Berkeley, CA, USA. [6]School of Biological Sciences, Nanyang Technological University, Singapore, Singapore. [7]Department of Biology, University of Naples Federico II, Naples, Italy. [8]Joint BioEnergy Institute, Lawrence Berkeley National Laboratory, Emeryville, CA, USA. [9]Biophysics, University of California Berkeley, Berkeley, CA, USA. [10]California Institute for Quantitative Biosciences (QB3), University of California Berkeley, Berkeley, CA, USA. [11]Center for Computational Biology, University of California Berkeley, Berkeley, CA, USA. [12]Division of Biology and Biological Engineering, California Institute of Technology, Pasadena, CA, USA. [13]School of Medicine, University of California San Diego, La Jolla, CA, USA. [14]Environmental Genomics and Systems Biology Division, Lawrence Berkeley National Laboratory, Berkeley, CA, USA. [15]Feedstocks Division, Joint BioEnergy Institute, Emeryville, CA, USA. [16]Department of Plant and Environmental Sciences, Weizmann Institute of Science, Rehovot, Israel. ✉e-mail: savage@berkeley.edu

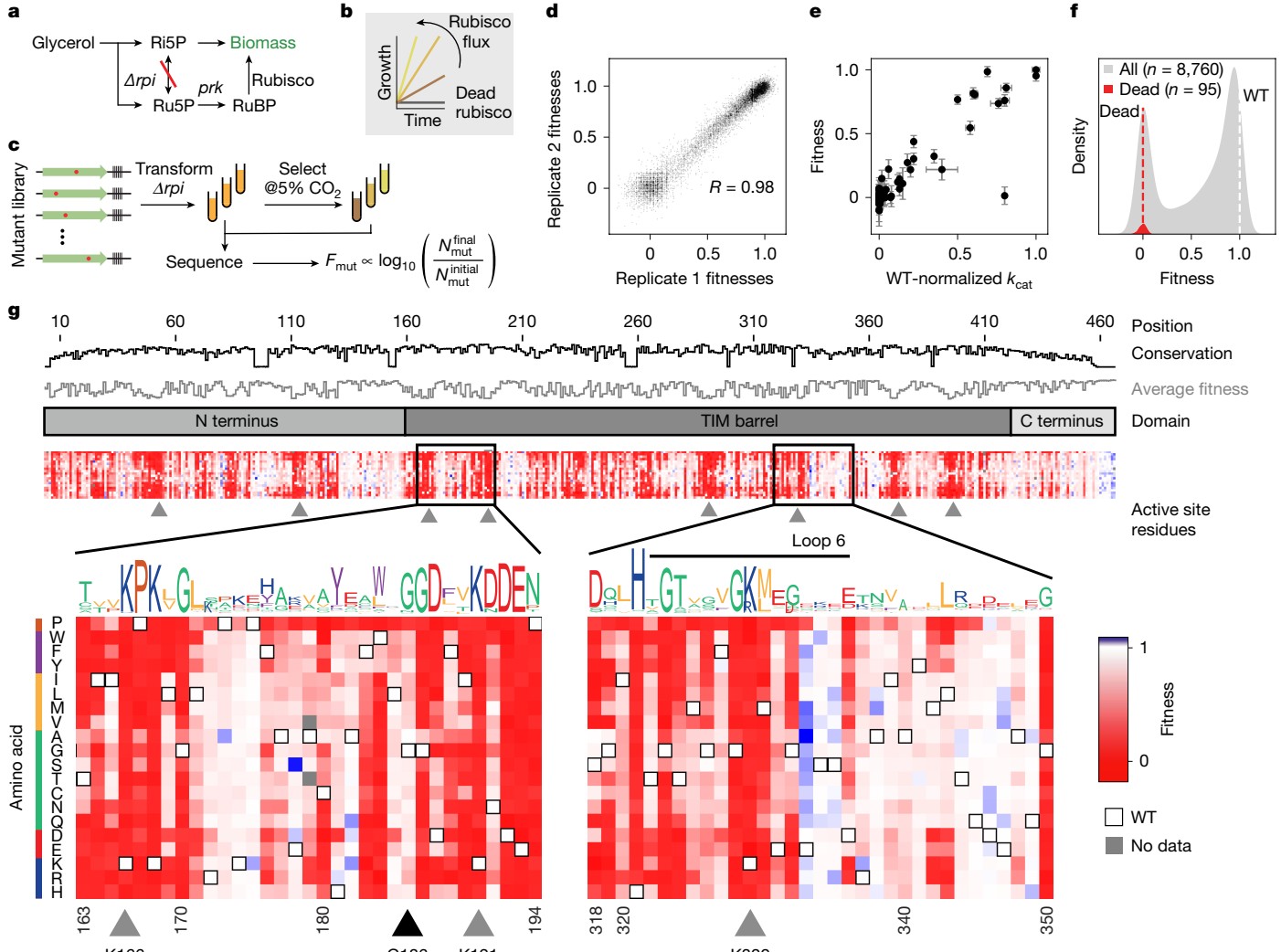

**Fig. 1 | A deep mutational scan individually characterizes all single-amino acid mutations in rubisco. a**, Summary of the metabolism of $\Delta rpi$—the rubisco-dependent strain. **b**, $\Delta rpi$ grows with a rate proportional to the flux through rubisco. **c**, Schematic of library selection. A library of rubisco single-amino acid mutants was transformed into $\Delta rpi$ then selected in minimal medium supplemented with glycerol at elevated $CO_2$. Samples were sequenced before and after selection and barcode counts were used to determine the relative fitness of each mutant. **d**, Correspondence between two example biological replicates; each point represents the median fitness among all barcodes for a given mutant. **e**, Fitness of 77 mutants with measurements in previous studies compared with the rate constants measured in those studies ($k_{cat}$). The outlier is I190T (see Methods for discussion). Fitness error values are the s.e.m. of nine replicate enrichment measurements; $k_{cat}$ errors are from the literature, where available. **f**, Variant fitnesses (grey) were normalized between values of 0 and 1, with 0 representing the average of fitnesses of mutations at a panel of known active site positions (red distribution, average is plotted as a red dashed line) and 1 representing the average of wild-type (WT) barcodes (white dashed line). **g**, Heatmap of variant fitnesses. Conservation by position and sequence logo were determined from a MSA of all rubiscos. Black triangle, G186 (an example of a position with high conservation that is mutationally tolerant); grey triangles, active site positions. Ri5P, ribose 5-phosphate; Ru5P, ribulose-5-phosphate; RuBP, ribulose-1,5-bisphosphate; TIM, triosephosphate isomerase.

rubisco rescue growth by converting this otherwise dead-end metabolite into 3-phosphoglycerate, which can feed back into central carbon metabolism (Fig. 1a and Extended Data Fig. 2a; for similar selection systems see refs. 23,24).

We first confirmed that the growth rate of $\Delta rpi$ was related quantitatively to known in vitro enzyme behaviour (Fig. 1b and Extended Data Fig. 2b–l). Expression of rubisco driven by an inducible promoter demonstrated that growth rates increased with the rubisco concentration, indicating that increased enzyme concentration led to higher fitness (Extended Data Fig. 2b,d,g); at isopropyl-β-D-thiogalactopyranoside (IPTG) concentrations above 30 μM, growth yields began to decline, indicating that rubisco overexpression comes with a fitness cost. Similarly, we observed faster growth in the presence of higher $CO_2$ concentrations (Extended Data Fig. 2c,d). We next assessed whether growth-based selection correlated with biochemical behaviour. Previous work on *R. rubrum* rubisco identified 77 mutants spanning from

less than 1% to 100% of wild-type catalytic rate (Supplementary Data 1). Growth of a subset of these mutants was tested and found to correlate with reported catalytic rates (Extended Data Fig. 2i–k). Together, these results are consistent with glycerol growth of $\Delta rpi$ being limited by rubisco carboxylation flux, which is determined by enzyme kinetics— $k_{cat}$ and $K_C$—as well as enzyme and $CO_2$ concentrations.

We next constructed a library of all single-amino acid substitutions to the model Form II rubisco from *R. rubrum* (Extended Data Fig. 3a). This library was cloned into a selection plasmid containing *PRK*, barcoded and bottlenecked to around 500,000 colonies. Long-read sequencing was used to map barcodes to mutants (Extended Data Figs. 3b and 4) and determined that the final library contained approximately 180,000 barcodes, representing 8,760 mutants or more than 99% of the designed library (Extended Data Fig. 4).

This library was transformed into $\Delta rpi$ to assess mutant fitness (Fig. 1c). Mutant fitness is defined by the relative growth rate of $\Delta rpi$

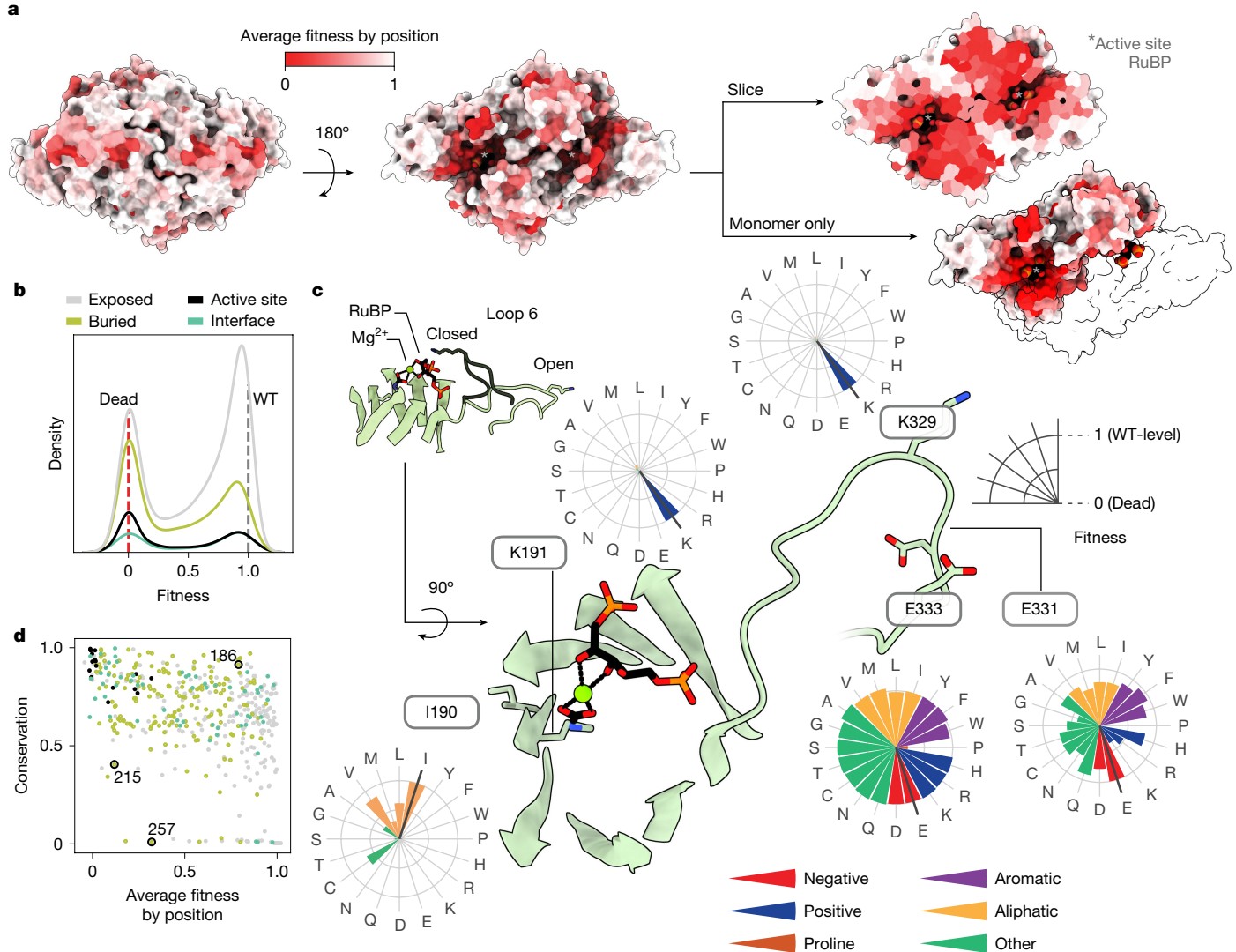

**Fig. 2 | Fitness values provide structural, functional and evolutionary insights into rubisco. a**, Structure of *R. rubrum* rubisco homodimer (Protein Data Bank (PDB) 9RUB) coloured by the average fitness value of a substitution at every site. Asterisks denote active sites. **b**, Variant effects for amino acids in different parts of the homodimer complex. **c**, Close-up view of the active site and the mobile Loop 6 region. Radar plots show the fitness effects of all mutations at a given position. **d**, Comparison of average fitness at each position against phylogenetic conservation among all rubiscos. Positions coloured as in **b**. Positions 215 and 257 form a tertiary interaction (Extended Data Fig. 8c), position 186 is highly conserved with no known function.

expressing that mutant. Three independent library transformations were grown in selective conditions and grown for around seven divisions in 5% $CO_2$ (equivalent to approximately 1,200 μM $CO_2$ in solution; wild-type $K_C$ = 150 μM). Selection was in the presence of 20 μM IPTG—a concentration at which rubisco is limiting and overexpression stress is minimized but growth is relatively robust (Extended Data Fig. 2b,d). Short-read sequencing quantified barcode abundance before and after selection (Methods). Mutant fitness was calculated by normalizing pre- and post-selection $\log_{10}$ read-count ratios to a panel of known catalytically dead mutants and all wild-type barcodes (Methods). Nine replicate experiments were performed with an average pairwise Pearson coefficient of 0.98 (Fig. 1d and Extended Data Fig. 5).

We compared mutant fitness measurements against 77 catalytic rate values taken from the literature (Fig. 1e and Supplementary Data 1), as well as 35 in vitro measurements from purified mutants (Extended Data Fig. 6a,b), and observed a linear relationship. Overall, we observed a bimodal distribution of mutant effects (Fig. 1f), with mutant fitnesses clustering near wild-type (neutral mutations) and catalytically dead variants[18,25].

We measured fitness values for more than 99% (8,760 out of 8,835) of amino acid substitutions (Fig. 1g and Extended Data Figs. 4f and 7b). Fewer than 0.14% of mutations seemed more fit than wild type (and when they did it was by a small amount (Fig. 1f)) and 72% were found to be deleterious. In vitro analysis of 11 variants with improved fitness did not show higher $k_{cat}$ values (Extended Data Fig. 6b) indicating that those small fitness effects were probably related to protein expression (Extended Data Fig. 2f–h). Mutations at known active site positions had very low fitness (for example, K191, K166 and K329; residues with grey triangles in Fig. 1g, bottom), and mutations to proline were more deleterious on average than any other amino acid (Extended Data Fig. 7a). Phylogenetic conservation and average fitness at each position tended to anti-correlate (Figs. 1g (top tracks) and 2d and Extended Data Fig. 8a) consistent with previous studies[26,27]; however, several positions seemed to be both highly conserved and mutationally tolerant (Fig. 1g, black triangle).

## Fitness variation across the structure

Our fitness assays showed that some regions of the rubisco structure are much more sensitive to mutation than others (Fig. 2a,b).

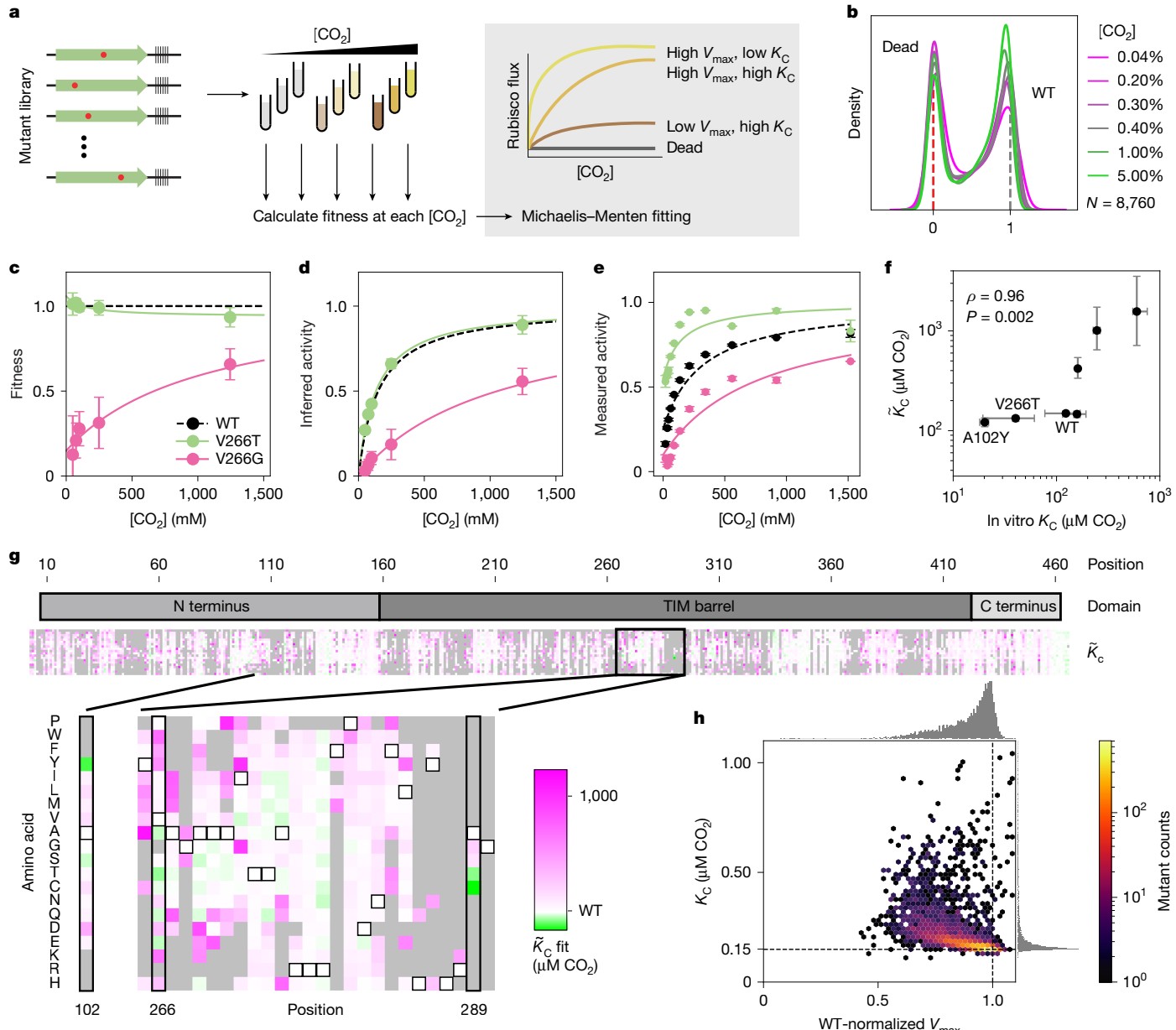

**Fig. 3 | $\widetilde{K}_C$ and $\widetilde{V}_{max}$ can be inferred from fitness across a $CO_2$ titration.**
**a**, Schematic of rubisco selection in $[CO_2]$ titration and some examples of inferred Michaelis–Menten curves of mutants with varying $K_C$ and $V_{max}$. **b**, Variant fitnesses at different $[CO_2]$. **c**, Measured fitnesses at different $[CO_2]$ for two mutants (error bars, s.d. of the mean for $N = 3$ biological replicates). **d**, The same data as in **c** plotted under the assumptions of the Michaelis–Menten equation (error bars, s.d. of the mean for $N = 3$ biological replicates). **e**, Individually measured rubisco kinetics for the same two mutants from **c** and **d** (points, medians of $N = 3$ measurements; error bars, s.d.). **f**, Comparison between rubisco $K_C$ values measured in vitro (spectrophotometric assay) and those

inferred from fitness values ($\widetilde{K}_C$). $\rho$ is calculated from a Spearman correlation; $P$ value reflects the result of a two-sided permutations test analysis. $\widetilde{K}_C$ error bars, inner quartiles of the bootstrap fits (Methods); in vitro $K_C$ error bars, s.d. from $N = 3$ measurements. **g**, Heatmap of $\widetilde{K}_C$ values for all mutants for which the coefficient of variation is less than 1 ($N = 5,687$ mutants, 65% of total). Two positions with high-affinity mutations are highlighted in the inset expanded below. Variants for which the $\widetilde{K}_C$ fits had a coefficient of variation above 1 are in grey. **h**, Two-dimensional histogram of mutant $\widetilde{K}_C$ and $\widetilde{V}_{max}$ values from **g** with hexagonal bins. Dashed lines, WT values.

For example, residues on the solvent-exposed faces of the structure are more tolerant to mutation, as expected, whereas active site and buried residues typically do not tolerate mutations well. A notable region of interest is Loop 6 of the triosephosphate isomerase barrel, which is known to fold over the active site during substrate binding and to participate in catalysis (Fig. 2c (inset) and Extended Data Fig. 1 (right panel)). Despite this key role in catalysis, some residues in this loop are highly tolerant to mutation (for example, E331 and E333), although the active site residue K329 is highly sensitive (Fig. 2c).

We expected that conserved positions would not tolerate mutations well. Consistent with this common hypothesis, the average fitness value at each position was negatively correlated with sequence conservation (Fig. 2d and Extended Data Fig. 8a). There were, however, many outliers, with several positions being highly conserved yet showing high mutational tolerance (for example, G186 (Fig. 2d, top right corner)). Selection in alternative conditions may reveal which selective forces have maintained high conservation at those positions[28]. Positions with low conservation and low mutational tolerance may indicate a recently evolved, but critical, function[26,27]; for example, M215 and H257

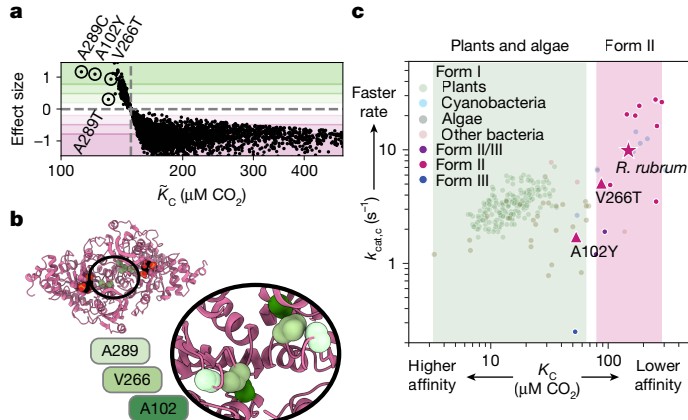

**Fig. 4 | Single-amino acid mutations can traverse the functional landscape.**
**a**, $\widetilde{K}_C$ versus effect size for each mutant. Effect size is the difference between the mutant $\widetilde{K}_C$ and WT $K_C$ divided by the coefficient of variation of $\widetilde{K}_C$. **b**, PDB structure 9RUB; inset on the $C_2$ symmetry axis is expanded below. Each position appears twice due to proximity to the $C_2$ axis. **c**, $k_{cat}$ versus $K_C$ of the indicated mutants (as measured by [14]C assay) versus all measured rubiscos from refs. 6,12. Shaded regions indicate known ranges of $\widetilde{K}_C$ values for plants and algae in green and Form II bacterial rubiscos in pink. Star, WT *R. rubrum*; triangles, mutants A102Y and V266T.

(Fig. 2d) are in contact in the *R. rubrum* structure but are absent in Form I sequences (Extended Data Fig. 8a–c).

## Affinity inferred by substrate titration

Enzyme fitness is determined by the underlying biochemical parameters, including catalytic rates and affinities. To measure these parameters individually, we performed a substrate titration on the whole library of mutations in tandem (Fig. 3a). Mutant fitness values varied overall with increasing $[CO_2]$ (Fig. 3b) and some mutants' fitnesses were affected strongly (Fig. 3c). We fit the data to a Michaelis–Menten model of catalysis to estimate effective maximum rates ($\widetilde{V}_{max}$) and $CO_2$ half-saturation constants ($\widetilde{K}_C$)[20] (the tildes distinguish library-derived fit parameters from those measured in vitro). This fitting (Fig. 3d; Methods) generated $\widetilde{V}_{max}$ and $\widetilde{K}_C$ estimates for every mutant (Fig. 3g and Extended Data Fig. 8c,d). We judged the reliability of the estimates by the coefficient of variation (s.d. over the mean; $\sigma/\mu$) of 1,100 bootstrap fits of the data for each mutation (Methods); we focus here on the 65% of the mutants (5,687) that had a coefficient of variation under 1 (ref. 26). The remaining 35% are primarily mutants with low fitness values (Extended Data Fig. 6e) that may fail to fold altogether, although at higher expression levels or in combination with other mutations it may yet be possible to produce reliable estimates of their effects on rate and affinity.

We validated our $\widetilde{K}_C$ estimates by purifying a set of seven mutants chosen to span a range of predicted $\widetilde{K}_C$ values and measuring their $CO_2$ affinities in vitro (Fig. 3e,f). Unexpectedly, for several mutants, the $K_C$ values measured in vitro were substantially lower (higher affinity) than expected from our previous estimates on the basis of fitness data. For example, the $\widetilde{K}_C$ of V266T was around 130 μM, but $K_C$ was determined to be roughly 80 μM $CO_2$ (Fig. 3f,g; highlighted box). Four mutations stood out in our analysis for having especially low $\widetilde{K}_C$: A102Y, V266T, A289C and A289T (Fig. 4a).

Our estimates of $\widetilde{V}_{max}$ correlated with fitness ($r = 0.93$; Extended Data Fig. 6h), indicating that it is the primary driver of rubisco flux. However, $V_{max} = k_{cat} \times$ [rubisco] so variation in $V_{max}$ can have two potential causes: rubisco expression level and $k_{cat}$. $\widetilde{V}_{max}$ estimates report the product of those two factors.

We further found that $\widetilde{V}_{max}$ and $\widetilde{K}_C$ estimates anti-correlate for variants with near-wild-type kinetics where the estimates are most

reliable (Fig. 3h). This correlation implies that, in the absence of selective pressure, most single-amino acid mutations impair $CO_2$ affinity and $V_{max}$ in tandem. It is important to note that, since the $CO_2$ addition step in catalysis is thought to be irreversible[29] and there is no binding site for $CO_2$ in the enzyme[30], all measured affinities reflect $CO_2$ on-rates. The observed anticorrelation between $\widetilde{V}_{max}$ and $\widetilde{K}_C$ may therefore be related to subtle changes in the electronics of the active site or the geometry of the bound sugar substrate before or during bond formation with $CO_2$. It is also possible that these effects are caused by changes to enzyme stability.

Mutations at three positions (A289C, A102Y, V266T, A289T) induced strong improvements in $CO_2$ affinity in vivo (Figs. 3g and 4a). Other mutations at these same positions reduced affinity (for example, V266G, A102F and A289G; Fig. 3c–g). These three positions are not part of the active site and sit near the $C_2$ axis of the rubisco homodimer interface (Fig. 4b). In this region of the structure, residues are in closest proximity to 'themselves', that is, to their counterpart residue in the other monomer of the homodimer. The role these amino acids play in $CO_2$ entry into the active site, active site conformation or electrostatics remains unclear.

In vitro measurements confirmed that V266T and A102Y possess improved $CO_2$ affinities (we were unable to purify A289C). This correspondence between $\widetilde{K}_C$ measured in vivo and $K_C$ measured in vitro stands in contrast to mutations with $\widetilde{V}_{max}$, where follow-up biochemistry (Extended Data Fig. 8b and Supplementary Data File 1) did not show faster $k_{cat}$ values. Variants with improved $\widetilde{V}_{max}$ were probably improved through higher protein expression. Unlike $V_{max}$, the affinity parameter is independent of enzyme concentration so $\widetilde{K}_C$ predictions are expected to be more accurate. V266T and A102Y both exhibit roughly proportional reductions in catalytic rate (Fig. 4c, Extended Data Fig. 9c and Extended Data Table 2). These mutations had no effect on $CO_2$ versus $O_2$ specificity (Extended Data Fig. 9a,c and Extended Data Table 2) indicating that the 'cost' of improved affinity is paid for in catalytic rate alone. A102Y had a reduced $K_{M,RuBP}$, whereas that of V266T did not change from wild type. It is unclear what relationship, if any, there is in the shifts in $K_C$ and $K_{M,RuBP}$. Overall, the $k_{cat}$ and $K_C$ measurements place these mutants outside the range heretofore measured among bacterial Form II variants and at the edge of the distribution of plants and algae.

## Conclusion

Among the narrow range of sequences measured here, it was possible to identify mutants with substantially improved $CO_2$ affinity, indicating that the enzyme parameter landscape is rugged, with apparent gain-of-function readily accessible. Form I plant rubiscos typically share less than 50% identity with Form II bacterial rubiscos (more than 200 mutations; Extended Data Fig. 8d) and are thought to have evolved under a different set of selective constraints. Furthermore, Form I and II rubiscos have different oligomeric states and Form II rubiscos lack the small subunit characteristic of Form I, so it is surprising that it is possible to traverse the functional space between them with just one amino acid change.

In this study, we were unable to account for two factors of metabolic flux through rubisco: protein expression and side-reactivity with oxygen. Fitness correlation with known $k_{cat}$ values (Fig. 1e and Extended Data Fig. 6a) and our in vitro measurements (Extended Data Fig. 6b) indicate that the data are predictive, even without knowledge of expression. However, mutations such as I164T cause differences in protein expression as a function of IPTG induction (Extended Data Fig. 2f,h) which has an effect on the relative growth rate as compared with wild type (Extended Data Fig. 2g). Indeed, when we examined mutations with fitness values higher than those of wild type, we observed a consistent regression in their $k_{cat}$ rates measured in vitro (Extended Data Fig. 6b, inset). We interpret this trend to indicate that some fraction of mutations have a small or no effect on $k_{cat}$ while modestly improving

expression levels. Further work is required to measure and account for this effect[16]. The side reaction of rubisco is also important, as increasing the oxygen concentration from 10% to 20% causes Δrpi to decline in growth rate and yield (Extended Data Fig. 2e), presumably because of 2-phosphoglycolate production. The effect of oxygen on individual mutants may be determined through an oxygen titration and library selection.

In *R. rubrum*, the present-day sequence evolved under constraints that include endogenous regulation, environmental selective pressure and possible trade-offs between enzymatic parameters. Various trade-offs have been proposed in the catalytic mechanism of rubisco[10,12], including one between catalytic rate and $CO_2$ affinity[11]. The reductions in $k_{cat}$ (but not $S_{C/O}$) observed in the mutants with the highest $CO_2$ affinity is consistent with such a trade-off (Fig. 4c). A selection of a library of higher order mutants that spans a wider range of rubisco functional possibilities could confirm or reject a trade-off. The trade-offs in bacterial rubiscos may also constrain the evolution of plant rubiscos. However, previous work comparing the sequence-to-function map of related proteins found substantial context dependence on the effects of mutations[18]. Due to advancements in expressing plant rubiscos in *E. coli*[31], it may be possible to use this assay to understand the biochemical constraints of the organisms responsible for nearly all of terrestrial photosynthesis[1].

The overall space of rubiscos remains largely unexplored, raising the question of whether natural evolution has already produced rubiscos optimized for every environment. Δrpi may permit a higher throughput exploration of sequence space to find regions that are constrained by different trade-offs and produce substantial engineering improvements.

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

# Methods

## Strains

We cloned in a combination of *E. coli* TOP10 cells, DH5α and NEB Turbo cells. Protein expression was carried out using BL21(DE3). *Δrpi* was produced previously[7] from the BW25113 strain. An *rpiB* knockout was obtained from the Keio collection. *rpiA* and the *edd* gene were knocked out through P1 transduction and subsequent curing of the kanamycin marker with pCP20 (ref. 32). The result of these three knockouts, ΔrpiABΔedd, was *Δrpi*. The EDD deletion makes the strain rubisco dependent when grown on gluconate—a feature we did not make use of in this study.

## Plasmids

Sequences and further details about plasmids used in this study can be found in Supplementary Data 3.

**pUC19_rbcL.** The rubisco mutant library was assembled in a standard pUC19 vector. This plasmid was used as a PCR template for each of the 11 sub-library ligation destination sites.

**NP-11-64-1.** Selections were conducted using a plasmid designed for this study with a p15 origin, chloramphenicol resistance, LacI controlling rubisco expression, TetR controlling PRK expression and a barcode.

**NP-11-63.** Protein overexpression in BL21(DE3) cells was conducted using pET28 with a SUMO domain upstream of the expressed gene[6]. pSF1389 is the plasmid that expresses the necessary SUMOlase, bdSENP1, from *Brachypodium distachyon*.

## Primers

All primers were purchased from IDT and the oligonucleotide pool was purchased from Twist Bioscience. For sequences, see Supplementary Data 3.

## Library design and construction

The *R. rubrum* rubisco sequence was codon-optimized for *E. coli* and mutated systematically by means of the scheme outlined in Extended Data Fig. 3. The rubisco gene was split into 11 pieces. For each of those pieces (around 200 base pairs (bp) each) all point mutants were designed and synthesized as oligonucleotide pools. Eleven oligo sub-library pools, containing all single mutants within their respective region of around 200 bp, were purchased from Twist Bioscience and each sub-library was amplified individually using Kapa Hifi polymerase with a cycle number of 15. Each rubisco gene fragment was inserted into a corresponding linearized pUC19 destination vector, containing the remainder of the rubisco sequence flanking the insert, through golden gate assembly. This assembly generated 11 sub-libraries of the full-length *R. rubrum* rubisco gene, with each sub-library containing a region of approximately 200 bp including all single mutants. Each of these 11 rubisco libraries were transformed separately into *E. coli* TOP10 cells and, in each case, more than 10,000 transformants were scraped from agar plates to ensure oversampling of the roughly 1,000 variants in each sub-library. Plasmids were purified from each sub-library and mixed together at equal molar ratios to generate the full protein sequence library.

To produce the final library for assay, a selection plasmid containing an induction system for rubisco and PRK (Tac- and Tet-inducible, respectively) was amplified with primers that included a random 30 nucleotide barcode. The linearized plasmid amplicon and the library were cut with BsaI and BsmBI, respectively, ligated together and transformed into TOP10 cells. Plasmid was purified by scraping around 500,000 colonies and transformed in triplicate into *Δrpi* cells. These transformations were grown in 2× yeast extract tryptone medium to log phase (optical density (OD) = 0.6) and frozen as 25% glycerol stocks.

## Bacterial growth analysis

Bacterial strains were grown overnight in 2× yeast extract tryptone medium to saturation and then backdiluted. Once cultures reached exponential growth ($0.3 < OD_{600} < 0.8$) they were diluted into 150 µl of M9 media in 96-well plates with 25 µg ml$^{-1}$ chloramphenicol and the indicated concentrations of anhydrotetracycline and IPTG to a final $OD_{600}$ of 0.005 or 0.0005. Growth was monitored in a Spark plate reader (Tecan) while maintaining 37 °C and the indicated $O_2$ and $CO_2$ concentrations. Shaking consisted of alternating 5 min of orbital and 5 min of double orbital modes and measurements were collected every 10 min. Growth yields were calculated up to 40 h and growth rates were calculated as the growth rate between $OD_{600}$ values of 0.001 and 0.01 (the most consistently exponential range in our curves).

## Long-read sequencing analysis

The plasmid library was cut with SacII and sent for Sequel II PacBio sequencing. Reads were aligned and grouped by their barcodes. All reads of a given barcode were aligned and a consensus sequence was obtained using SAMtools[33]. Consensus sequences were retained if they were WT or had one mutation that matched the designed library. Any mutation in the backbone invalidated a barcode. A lookup table was generated to link each barcode to its associated mutation. The data analysis methods described in this study are publicly available at https://github.com/SavageLab/rubiscodms.

## Library characterization and screening

Selections were performed by diluting 200 µl of glycerol stock with an OD of around 0.25 into 5 ml of M9 minimal medium with added chloramphenicol (25 µg ml$^{-1}$), glycerol (0.4%), 20 µM IPTG and 20 nM anhydrotetracycline. These cultures were grown in 11 ml culture tubes at 37 °C in a Percival AR-22 growth chamber at different $CO_2$ concentrations on a New Brunswick Scientific Innova 2000 shaker at 250 rpm at an angle of 60°. Cultures were grown until they reached an OD at 5 ml of 1.2 ± 0.2. This corresponds to a 100-fold expansion of the cells, that is, between six and seven doublings.

Cultures before and after selection were spun down and we lysed the cells and performed a standard plasmid extraction protocol using QIAprep Spin Miniprep Kit (Qiagen). Illumina amplicons were generated by PCR of the barcode region. These amplicons were sequenced using a NextSeq P3 kit.

## Calculation of variant enrichment

Variant enrichments were computed from the log ratio of barcode read counts. The enrichment calculations include two processing parameters: a minimum count threshold ($c_{min}$) and a pseudo-count constant ($\alpha_p$). The count threshold is the minimum number of barcode reads that must be observed either pre- or post-selection for the barcode to be included in the enrichment calculation. The pseudo-count constant is used to add a small positive value to each barcode count to circumvent division by zero errors. We use a pseudo-count value that is weighted by the total number of reads in each condition. For the *j*th variant and the individual barcodes, *i*, passing the threshold condition the variant enrichment is calculated as,

$$e_j = \text{median}\left( \log_{10}\left( \frac{N_{f,i} + N_{f,\text{tot}}\alpha_p}{N_{0,i} + N_{0,\text{tot}}\alpha_p} \right) - \log_{10}\left( \frac{N_{f,\text{tot}}}{N_{0,\text{tot}}} \right) \right) \tag{1}$$

To identify optimal values for these parameters, we computed the variant enrichments across a two-dimensional parameter sweep of $c_{min}$ and $\alpha_p$ to find the combination that resulted in the maximum mean Pearson correlation coefficient across all replicates at each condition. These were $c_{min} = 5$ and $\alpha_p = 3.65 \times 10^{-7}$ (average of 0.3 pseudo-counts after multiplying by the total number of reads in each experiment, $N_{0,\text{tot}}$ or $N_{f,\text{tot}}$), leading to a correlation coefficient of 0.978.

Variant enrichment, $e_j$, was then calculated for every mutant using equation (1).

The variant enrichments were then normalized such that wild type has an enrichment value of 1 in all conditions and catalytically dead mutants have a median enrichment of 0. For the 'dead' variant enrichment, we computed the median enrichment for all mutations at the catalytic positions K191, K166, K329, D193, E194 and H287. The normalized enrichments at each condition were computed as

$$e_{j,\text{norm}} = \frac{e_j - \tilde{e}_{\text{dead}}}{e_{\text{WT}} - \tilde{e}_{\text{dead}}} \qquad (2)$$

where $e_j$ is the enrichment of the $j$th variant as given in equation (1), $e_{\text{WT}}$ is the wild-type enrichment and $\tilde{e}_{\text{dead}}$ is the median enrichment across all mutants of the catalytic residues listed above.

## Michaelis–Menten fits to enrichment data

The DMS library enrichments across different $CO_2$ concentrations were used to estimate Michaelis–Menten kinetic parameters for every variant. Guided by the linear relationship between growth rate and $k_{\text{cat}}$ observed in Fig. 1e, we assume that the cell growth rate is proportional to the rubisco enzyme velocity to derive the $CO_2$ titration fits (see 'Derivation of Michaelis–Menten fit', equation (S1))

$$e_{\text{mut,norm}}([CO_2]) = \frac{\widetilde{V}_{\text{max,mut}}(\widetilde{K}_{C,\text{WT}} + [CO_2])}{\widetilde{V}_{\text{max,WT}}(\widetilde{K}_{C,\text{mut}} + [CO_2])} \qquad (3)$$

$\widetilde{V}_{\text{max,mut}}/\widetilde{V}_{\text{max,WT}}$ is the ratio of mutant maximum velocity relative to wild type, $\widetilde{K}_{C,\text{WT}}$ is the wild-type $K_C$ for which we used the value 149 µM, and $\widetilde{K}_{C,\text{mut}}$ is the mutant $K_C$. The titration curves in triplicate for each variant were fit to equation (3) using non-linear least squares curve fitting while requiring both $V_{\text{max}}$ and $K_{C,\text{mut}}$ to be positive.

We noted that the $\overline{K}_C$ fits to certain variants—particularly ones with low $\widetilde{V}_{\text{max}}$—were sensitive to the choice of processing parameters $c_{\text{min}}$ and $\alpha_p$. Given the semi-arbitrary nature of these parameters, this is clearly an undesirable dependence and engenders low confidence in the inferred $\overline{K}_C$ values. To account for this uncertainty we conducted a parameter sweep (with 11 different $c_{\text{min}}$ values linearly spaced between 0 and 50, and 10 $\alpha_p$ values log spaced between $1 \times 10^{-9}$ and $1 \times 10^{-6}$), and computed the variant enrichments for all combinations of these parameters. We then performed ten subsamplings of the replicates for all parameter sets and performed the ratiometric Michaelis–Menten fit. From this set of 1,100 $\overline{K}_C$ fit values for each variant we computed a quartile-based coefficient of variation that was used as a figure of merit for the $\overline{K}_C$.

## Multiple sequence alignment

A multiple sequence alignment (MSA) of the broader rubisco family beyond Form II rubiscos was created using the profile HMM homology search tool jackhmmer[34]. Starting with the *R. rubrum* rubisco sequence, jackhmmer applied five search iterations with a bit score threshold of 0.5 bits per residue against the UniRef100 database of non-redundant protein sequences[35]. To compute phylogenetic conservation at each position, for each possible amino acid we computed the fraction of the total sequences that had that amino acid at the corresponding position of the MSA. The phylogenetic conservation is the maximum fraction, where the maximum is taken over all possible amino acids. Thus, if a position has an alanine in 90% of the sequences of the MSA, the phylogenetic conservation will be 0.9.

## Protein purification

*E. coli* BL21(DE3) cells were transformed with pET28 (encoding the desired rubisco with a 14× His and SUMO affinity tag) and pGro plasmids (Takara). Colonies were grown at 37 °C in 100 ml of 2× yeast extract tryptone medium under kanamycin selection (50 µg ml⁻¹) to an OD of 0.3–1. Arabinose (1 mM) was added to each culture, which was then incubated at 16 °C for 30 min. Protein expression was induced with IPTG (Millipore) at 100 µM and cells were grown overnight at 18 °C. Cultures were spun down (15 min; 4,000g; 4 °C) and purified as reported[6]. In brief, cultures were spun down and lysed using BPER II (Thermo Fisher). Lysates were centrifuged to remove insoluble fraction. Rubisco was purified by His-tag purification using Ni-NTA resin (Thermo Fisher) and eluted by SUMO tag cleavage with bdSUMO protease (as produced in ref. 6). Purified proteins were concentrated and stored at 4 °C until kinetic measurement (within 24 h). Samples were resolved by SDS–polyacrylamide gel electrophoresis to ensure purity.

## Rubisco spectrophotometric assay

Both $k_{\text{cat}}$ and $K_C$ measurements use the same coupled-enzyme mixture wherein the phosphorylation and subsequent reduction of 1,3-bisphosphoglycerate—the product of RuBP carboxylation—was coupled to NADH oxidation, which can be followed through 340-nm absorbance. Following Kubien et al.[36] and Davidi et al.[6], the reaction mixture (Extended Data Table 1) contains buffer at 100 mM, pH 8, 20 mM $MgCl_2$, 0.5 mM dithiothreitol, 2 mM ATP, 10 mM creatine phosphate, 1.7 mM NADH, 1 mM EDTA and 20 U ml⁻¹ each of phosphoglycerate kinase, glyceraldehyde-3-phosphate dehydrogenase and creatine phosphokinase. Reaction volumes are 150 µl and samples are shaken once before absorbance measurements begin. Absorbance measurements are collected on a SPARK plate reader with $O_2$ and $CO_2$ control (Tecan). The extinction coefficient of NADH in the plate reader was determined through a standard curve of NADH solutions of known concentration (determined by a Genesys 20 spectrophotometer with a standard 1-cm path length, Thermo Fisher). Absorbance decline over time gives a rate of NADH oxidation and therefore a carboxylation rate. Because rubisco produces two molecules of 3-phosphoglycerate for every carboxylation reaction, we assume a 2:1 ratio of NADH oxidation rate to carboxylation rate.

**Spectrophotometric measurements of $k_{\text{cat}}$.** The carboxylation rate constant ($k_{\text{cat}}$) of each rubisco was measured using methods established previously[6]. In brief, rubisco was activated by incubation for 15 min at room temperature with $CO_2$ (4%) and $O_2$ (0.4%) and added (final concentration of 80 nM) to aliquots of appropriately diluted assay mix (Extended Data Table 1) containing different 2-carboxy-D-arabinitol-1,5-bisphosphate (CABP) concentrations pre-equilibrated in a plate reader (Infinite 200 PRO; TECAN) at 30 °C, under the same gas concentrations. After 15 min, RuBP (final concentration of 1 mM) was added to the reaction mix and the absorbance at 340 nm was measured to quantify the carboxylation rates. A linear regression model was used to plot reaction rates as a function of CABP concentration. The $k_{\text{cat}}$ was calculated by dividing the $y$ intercept (reaction rates) by the $x$ intercept (concentration of active sites). Protein was purified in triplicate for $k_{\text{cat}}$ determination.

**Spectrophotometric measurements of $K_C$.** Purified rubisco mutants were activated (40 mM bicarbonate and 20 mM $MgCl_2$) and added to a 96-well plate along with assay mix (Extended Data Table 1, in this case the same concentration of HEPES pH 8 buffer was used but EPPS can be substituted). Bicarbonate was added for a range of concentrations (1.5, 2.5, 4.2, 7, 11.6, 19.4, 32.4, 54, 90 and 150 mM). Plates and RuBP were pre-equilibrated at 0.3% $O_2$ and 0% $CO_2$ at room temperature. RuBP was added to a final concentration of 1.25 mM with water serving as a control for each replicate. NADH oxidation was measured by $A_{340}$ as in the $k_{\text{cat}}$ assay. Absorbance curves were analysed using a custom script to perform a hyper-parameter search to choose a square in which to take the slope as carboxylation rate that best represented most of the monotonic decrease in $A_{340}$. $K_C$ was derived by fitting the Michaelis–Menten curve using a non-linear least squares method. Error bars were determined depending on replicates: (1) multi day replicates: Michaelis–Menten

fits were made for each replicate, s.e. and median were calculated on the basis of these fits. (2) Triplicates: absorbance data were fit to extract initial rates using different hyper-parameters and the median of these fits was used subsequently. Three different sets of initial rates were calculated on the basis of the technical replicates: one based on the median absorbance values, one based on the median minus the s.d., and one based on the median plus the s.d. Michaelis–Menten fits to these three sets of calculated rates were made and error bars show the difference between the low boundary, median and high boundary set.

**Spectrophotometric measurements of $K_{M,RuBP}$.** $K_{M,RuBP}$ was determined in a similar manner to $k_{cat}$ and $K_C$. A titration of RuBP concentrations was used to generate rate-saturation curves under an atmosphere of 5% $CO_2$ and 0.5% $O_2$. Simple linear regression was used to fit the absorbance decays. $K_{M,RuBP}$ was derived by fitting the Michaelis–Menten curve using a non-linear least squares method. Error was determined from the square root of the diagonals of the covariance matrix during fitting. The values from spectrophotometric assays are reported in Fig. 3f and Extended Data Figs. 6b,d and 9b.

**Radiometric measurements of $K_C$ and $k_{cat}$.** [14]$CO_2$ fixation assays were conducted as in ref. 6 with minor modifications. Assay buffer (100 mM EPPS-NaOH pH 8, 20 mM $MgCl_2$, 1 mM EDTA) was sparged with $N_2$ gas. Rubisco, purified as described above, was diluted to around 10 μM (quantified using ultraviolet absorbance) in the assay buffer. It was then diluted with one volume of assay buffer containing 40 mM NaH[14]$CO_3$ to activate. Reactions (0.5 ml) were conducted at 25 °C in 7.7-ml septum-capped glass scintillation vials (Perkin-Elmer) with 100 μg ml[−1] carbonic anhydrase, 1 mM RuBP and NaH[14]$CO_3$ concentrations ranging from 0.4 to 17 mM (which corresponds to 15–215 μM $CO_2$). The assay was initiated by the addition of a 20-μl aliquot of activated rubisco and stopped after 2 min by the addition of 200 μl 50% (v/v) formic acid.

The specific activity of [14]$CO_2$ was measured by performing a 1-h assay at the highest [14]$CO_2$ concentration containing 10 nmol of RuBP. Reactions were dried on a heat block, resuspended in 1 ml water and mixed with 3 ml Ultima Gold XR scintillant for quantification with a Hidex scintillation counter.

The rubisco active site concentration used in each assay was quantified in duplicate by a [14C]-2-CABP binding assay. A 10-μl sample of the roughly 10 μM rubisco solution was activated in assay buffer containing 40 mM cold NaHCO₃ (final volume 100 μl) for at least 10 min. Then, 1.5 μl of 1.8 mM [14]C-carboxypentitol bisphosphate was added and incubated for at least 1 h at 25 °C. [14C]-2-CABP bound rubisco was separated from free [14C]-2-CPBP by size exclusion chromatography (Sephadex G-50 Fine, GE Healthcare) and quantified by scintillation counting.

The data were fit to the Michaelis–Menten equation using the concatenated data of three to four experiments performed on different days. This assay was used to determine the values in Fig. 4c and Extended Data Fig. 9c.

**Membrane inlet mass spectrometry determination of rubisco specificity.** The method described in ref. 37 was adapted for a membrane inlet mass spectrometry (MIMS) instrument (Bay Instruments). The $O_2$ ion signal was calibrated by measuring the 32 $m/z$ ion at atmospheric $O_2$ and at 'zero' $O_2$. An atmospheric $O_2$ calibrant was achieved by equilibrating the MIMS buffer (200 mM Hepes pH 8, 100 mM NaCl, 20 mM $MgCl_2$) with air for 1 h at 25 °C. The 'zero' $O_2$ ion signal was determined by then adding approximately 5 mg $Na_2S_2O_6$ to the cuvette. $CO_2$ was calibrated by adding various amounts of NaHCO₃ to a solution of 100 mM HCl and recording the 44 $m/z$ ion signal. In both cases, linear fits of ion counts to gas concentrations provided a simple conversion to determine gas concentrations and consumption rates. These calibrations had to be performed on every day in which the assay was used.

Rubisco enzymes were activated in 20 mM Hepes pH 8, 100 mM NaCl, 20 mM $MgCl_2$ and 20 mM NaHCO₃. Activated enzyme was added to 630 μl of MIMS buffer equilibrated with air at a concentration of 1.2 μM. Bovine carbonic anhydrase (Sigma Aldrich) was added at a final concentration of 0.3 mg ml[−1] and NaHCO₃ was added to a final concentration of 4 mM. The reaction was stirred in the sealed MIMS reaction chamber for approximately 2 min to collect a pre-reaction signal. The reaction was initiated by the addition of 2 mM RuBP. $O_2$ and $CO_2$ consumption rates were background corrected and converted to reaction velocities through conversion using the coefficients determined during calibration. Specificities were determined in triplicate by the following equation: $S_{C/O} = v_C[O_2]/v_O[DIC]$, where DIC is the dissolved inorganic carbon pool.

## Quantification of soluble enzyme concentration by immunoblotting

The $\Delta rpi$ strain with wild-type rubisco was grown under selective conditions (overnight at 37 °C in M9 medium with 0.4% glycerol and 20 nM aTc) with varying IPTG concentrations at 5% $CO_2$ for 24 h. Afterwards, turbid cultures were centrifuged (10 min; 4,000$g$; 4 °C) culminating in roughly 20 mg pellet per sample. Pellets were lysed with 200 μl of BPER II and supernatant was transferred into a fresh tube and mixed with SDS loading dye. A Bio-Rad RTA Transfer Kit for Trans-Blot Turbo Low Fluorescence PVDF was used in combination with the Trans-Blot Turbo Transfer System. The PVDF membrane was carefully cut between 50 and 70 kDa post-blocking using a razor blade. Primary anti-RbcL II Rubisco large subunit Form II Antibody from Agrisera (1:10,000) and DnaK Antibody from Abcam (1:5,000) were incubated separately. Secondary horseradish-peroxidase-conjugated antibodies Donkey anti-mouse for DnaK (Santa Cruz Biotechnology) and Goat pAB to RB IgG horseradish peroxidase (Abcam were both used at 1:10,000). Subsequently, Bio-Rad Clarity Max Western ECL substrates were applied and the final results were imaged using a GelDoc (Bio-Rad).

## Mutant fitness outlier

I190T (Fig. 1e) was the only outlier in our comparison of in vitro $k_{cat}$ measurements from the literature and our fitness data. Because the value was reported without error estimates[38], we re-measured the $k_{cat}$ of this mutant and found it to be 4.24 s[−1], which is 52% of the wild-type value, down from 80% previously reported. Still, the value seems to be anomalous compared with the rest of the trend (Extended Data Fig. 6b). One potential explanation is that the mutation at that position has a strong negative effect on protein expression. Another possibility, given that I190T is adjacent to the key active site lysine, K191, is that I190T causes a negative effect on lysine carbamylation that is, for some reason, more pronounced in vivo than in vitro.

## Derivation of Michaelis–Menten fit

Following Stiffler et al.[20] we assume that the differences in bacterial growth rate are proportional to the differences in growth-limiting enzymatic activity.

$$\mu_{mut} - \mu_{WT} \propto v_{mut}^{ru} - v_{WT}^{ru} \tag{S1}$$

Under the presumption of log-phase growth, the expected log ratio of reads after elapsed time $t$ and normalized to the wild-type reads is given by

$$e_{mut} = \log_{10}\left(\frac{N_{mut,f}}{N_{mut,0}}\right) - \log_{10}\left(\frac{N_{WT,f}}{N_{WT,0}}\right) \tag{S2}$$

(Note that equation (S2) would also contain a normalization factor to account for the total number of reads obtained for the pre- and post-selection conditions. It is, however, a common factor for both the mutant and wild-type counts and therefore cancels out. Furthermore, the real analysis also includes pseudo-counts, which are omitted here in

the derivation of the fit equation for simplicity. Substituting in the condition of exponential growth, that is, $N_{i,f} = N_{i,0} \, e^{\mu_i t}$, and simplifying yields,

$$e_{\text{mut}} = \frac{t}{\ln 10}(\mu_{\text{mut}} - \mu_{\text{WT}}) \tag{S3}$$

To normalize the enrichments, we divide by the log enrichment of the wild-type counts relative to the median enrichment of variants with mutated catalytic residues (and thus catalytically dead rubisco). We then add one for the convention that dead variants be centred at an enrichment of 0 and that wild-type be at an enrichment of 1. Thus, the normalized mutant enrichment is,

$$e_{\text{mut,norm}} = \frac{\log_{10}\left(\frac{N_{\text{mut},f}}{N_{\text{mut},0}}\right) - \log_{10}\left(\frac{N_{\text{WT},f}}{N_{\text{WT},0}}\right)}{\log_{10}\left(\frac{N_{\text{WT},f}}{N_{\text{WT},0}}\right) - \left\langle \log_{10}\left(\frac{N_{\text{dead},f}}{N_{\text{dead},0}}\right) \right\rangle} + 1 \tag{S4}$$

Then substituting equation (S3) we obtain,

$$e_{\text{mut,norm}} = \frac{\mu_{\text{mut}} - \mu_{\text{WT}}}{\mu_{\text{WT}} - \underline{\mu_{\text{dead}}}} + 1 \tag{S5}$$

Using the assumption in equation (S1) and the fact that the enzyme velocity of dead mutants is 0, we obtain the expected normalized enrichment as a function of the rubisco velocities,

$$e_{\text{mut,norm}} = \frac{\upsilon_{\text{mut}}}{\upsilon_{\text{WT}}} \tag{S6}$$

Finally, using the Michaelis–Menten equation we obtain the predicted enrichments as a function of $CO_2$ concentration and the enzyme kinetic parameters.

$$e_{\text{mut,norm}}([CO_2]) = \frac{V_{\text{max,mut}}(K_{M,\text{WT}} + [CO_2])}{V_{\text{max,WT}}(K_{M,\text{mut}} + [CO_2])} \tag{S7}$$

Thus, in practice, we use equation (S7) as the fit equation to the normalized enrichment values for each variant across a range of $CO_2$ concentrations. For each we have, as fit parameters, the ratio of maximum velocities between the mutant and wild type, $V_{\text{max,mut}}/V_{\text{max,wt}}$, and the mutant $K_C$ with the wild-type $K_C$ set to the literature value of 149 μM.

## Reporting summary

Further information on research design is available in the Nature Portfolio Reporting Summary linked to this article.

## Data availability

All data for this paper are available at https://github.com/SavageLab/rubiscodms. Sequences for our Form II rubisco phylogeny were assembled from UniRef100. Our raw sequencing reads can be accessed on the NCBI SRA (accession PRJNA1181558). All other data are available in the paper or the Supplementary Information.

## Code availability

All code for this paper is available at https://github.com/SavageLab/rubiscodms.

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

**Acknowledgements** We thank N. Antonovsky and A. Bar-Even for taking part in formulating the basis for this work, as well as N. Tepper and S. Amram for originally conceiving of and producing the *Δrpi* strain, respectively. We thank P. Romero, N. Thompson, L. Fedotov, O. Saltzman, E. Prywes, S. Wyman, B. Yu and J. Desmarais for essential help in the process of data analysis. For their assistance in the process of generating and validating the DMS library, we thank A. Glazer, K. Matreyek, J. Bloom and K. Reynolds. Additionally, we thank J. Tartaglia for the use of her sequencing primers and N. Krishnappa for assistance in running NGS samples. We would like to thank E. Meng for assistance using ChimeraX. Finally, we thank F. Wang for technical assistance over the weekends. D.F.S. is an Investigator of the Howard Hughes Medical Institute. This work was supported by US National Institutes of Health grant no. K99GM141455-01 (N.P.) and the US Department of Energy, Physical Biosciences Program, award number DE-SC0016240 (D.F.S.).

**Author contributions** Conceptualization: N.P., A.I.F., D.F.S. Methodology: N.P., N.R.P., L.M.O., L.J.T.-K., R.Z.W., S.L., D.D., O.M.-C., R.M., D.F.S. Investigation: N.P., N.R.P., S.L., Y.-C.C.T., B.d.P., A.E.C., L.J.T.-K., R.Z.W., H.A.C., L.N.H., D.B.-R., H.M.N., R.F.W., A.Y.B. Visualization: N.P., L.M.O., S.L., D.F.S. Funding acquisition: N.P., D.F.S. Project administration: N.P., D.F.S. Supervision: N.P., P.M.S., O.M.-C., R.M., D.F.S. Writing—original draft: N.P., L.N.H., A.I.F., D.F.S.

**Competing interests** D.F.S. is a co-founder and scientific advisory board member of Scribe Therapeutics. The other authors declare no competing interests.

**Additional information**
**Correspondence and requests for materials** should be addressed to David F. Savage.

**Extended Data Fig. 1 | *R. rubrum* rubisco structure.** Left, Overall structure of the 2-large subunit (L2) homodimer with active sites and $C_2$-symmetry axis labelled with a black two-fold axis symbol- ●. (PDB: 9RUB). Centre, Ribbon diagram of one monomer with the 3 subdomains labelled. View is of the interfacial side. Right, Close-up view of the active site. Closed form of loop 6 is from the 8RUC structure. Active site residues and RuBP substrate are labelled.

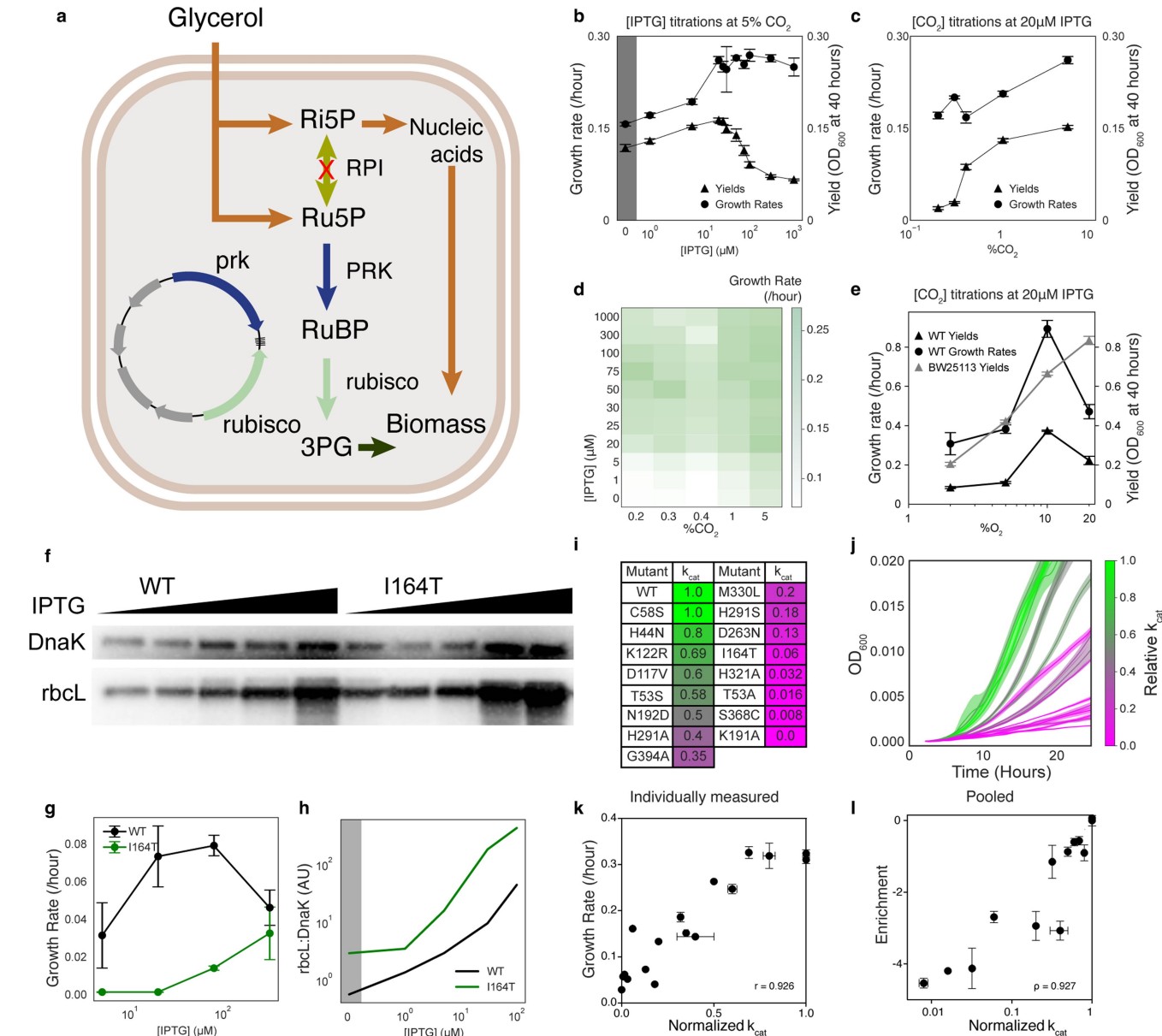

**Extended Data Fig. 2 | Δrpi is a rubisco-dependent E. coli strain with a growth rate that correlates to rubisco flux. a)** Schematic of the Δrpi strain of rubisco-dependent E. coli. PRK and rubisco compensate for the deletion of RPI and rescue growth. **b)** Growth rates and yields across a titration of rubisco induction by [IPTG]. (N = 4) **c)** Growth rates and yields across a titration of [CO₂]. Yields were calculated up to 40 h. (N = 4) **d)** A heatmap of growth rates across a two-dimensional titration of CO₂ and IPTG. **e)** Growth rates and yields across a titration of [O₂]. Yields were calculated between 15 and 40 h. The BW25113 contained the same plasmid as Δrpi but with GFP in place of rubisco. Growth rates could not be calculated for the control due to non-exponential growth behavior. (N = 6) **f)** Immunoblots for soluble rubisco with DnaK as a loading control. Left half is wild-type R. rubrum rubisco, right half is the higher-expressing I164T mutant. Samples are of Δrpi cells grown in selection media (see Methods) with different concentrations of IPTG. **g)** Growth rates of Δrpi cells expressing either WT or I164T rubisco grown in selection media with different concentrations of IPTG. (N = 4) **h)** Ratio of band intensities from **f** as a function of IPTG concentration. **i)** A panel of mutants from the literature and their associated $k_{cat}$ measurements normalised to WT. The WT value is ≈11/s. **j)** Growth curves of Δrpi expressing the mutants from **i**. Colouring in **i** and **j** is on the same scale and reflects $k_{cat}$ values from the literature. **k)** Growth rate values calculated from the curves in **j**, plotted against the normalised $k_{cat}$ values. **l)** Raw barcode-averaged mutant enrichment values for the same mutants as in **k** measured in one nanopore sequencing experiment. Error bars in **b, c, g** and **e** determined from the SEM of at least four replicates. Error bars in **k** determined as standard deviations of three or more replicates. Error bars in **l** determined as standard deviations of three different barcodes (N = 3) for each mutant. Errors in literature values are shown from studies where they were reported.

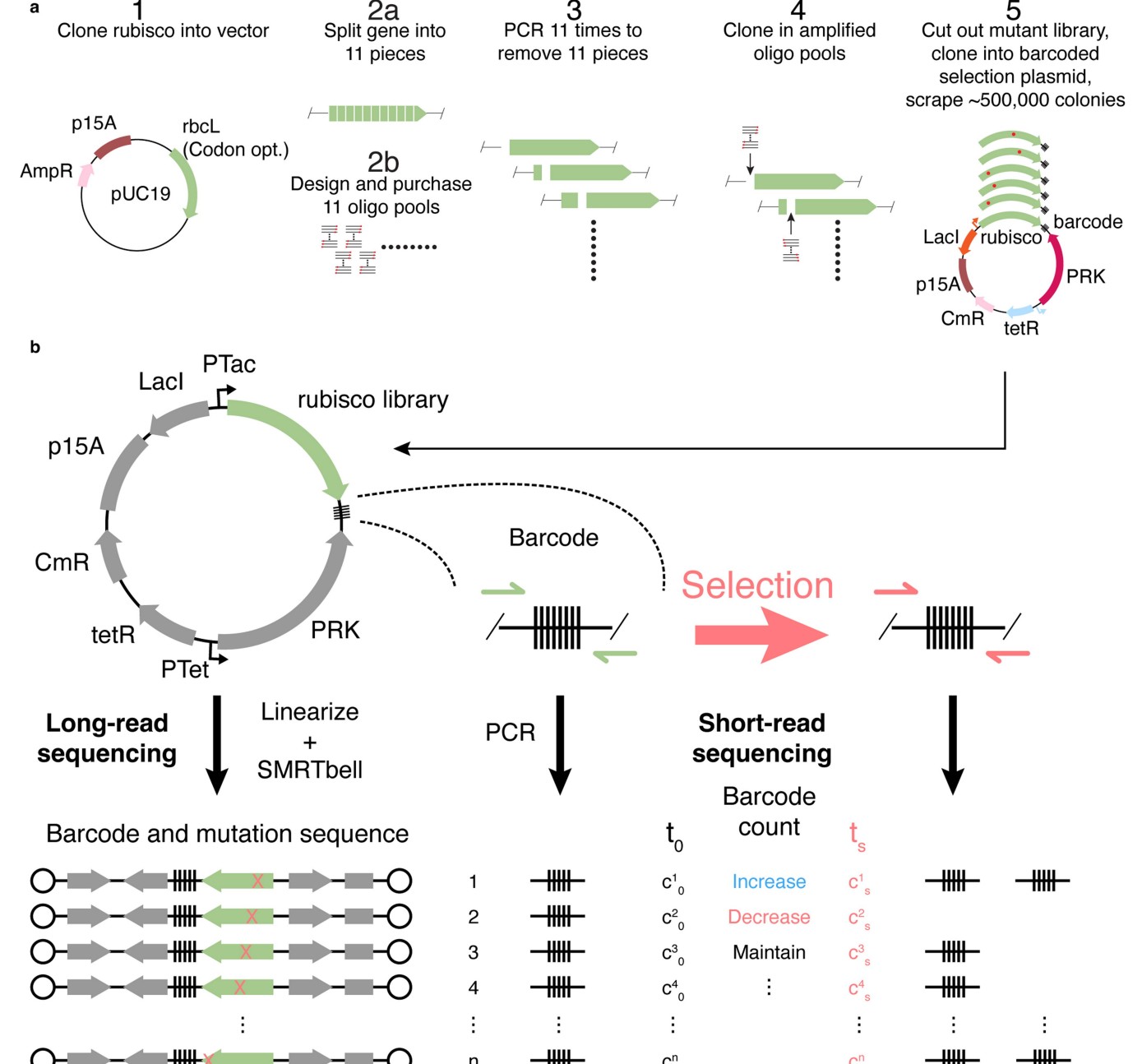

**Extended Data Fig. 3 | Library construction and characterization pipeline.**
**a)** Library construction procedure. **Step 1)** Clone a codon-optimised *R. rubrum* rubisco sequence into pUC19. **Step 2a)** Choose locations to split the gene which are appropriate for the cloning of subpool libraries. **Step 2b)** PCR amplify the sub-libraries from an oligo pool containing all 8778 mutations. **Step 3)** PCR amplify the backbone with a space missing for the ligation of an oligo subpool.

**Step 4)** Ligate each oligo subpool to its appropriate backbone. **Step 5)** Combine the sub libraries, cut the full, mutated genes out and ligate them into a PCR-amplified and barcoded backbone. After transformation scrape the desired number of colonies for selection. **b)** Library sequencing strategy. The library was characterised by long read sequencing. Barcode abundances were measured by short-read sequencing before and after selection (see methods).

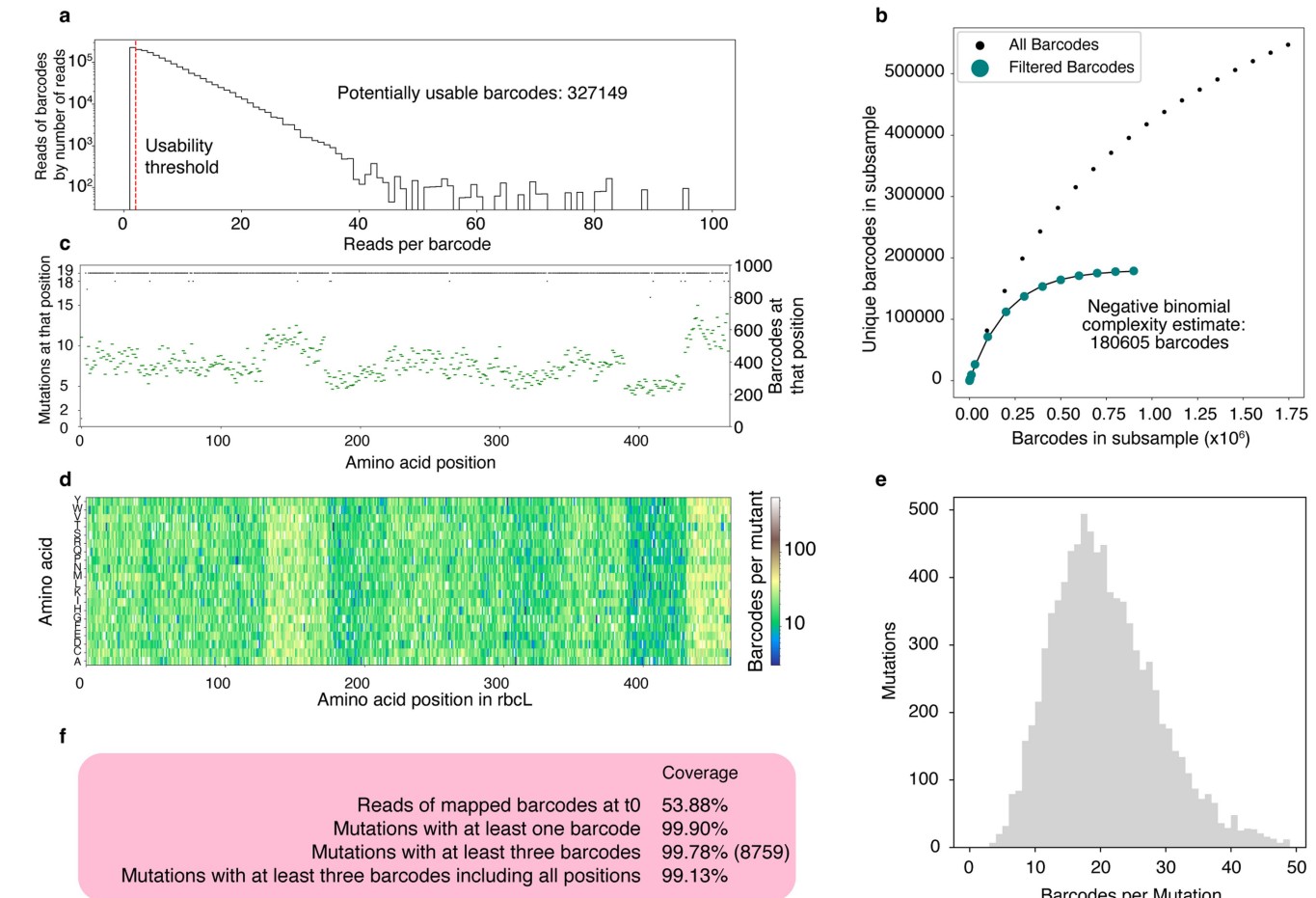

**Extended Data Fig. 4 | Library characterization by long-read sequencing.**
**a)** A histogram of reads of plasmids from PacBio sequencing. The y-axis represents the number of reads of plasmids with a given number of reads (i.e. the bar at 50 on the x-axis is as tall as the number of reads of barcodes with 50 reads). We were able to generate a consensus sequence for any barcode with more than 1 read leaving us with 327,149 possible barcodes. **b)** A rarefaction plot estimating the overall library complexity, a negative binomial distribution was fit and we estimated a real library complexity of ≈180,000 barcodes. **c)** A plot of how many mutants (of the possible 19) were in our library at each position (black dashes, left axis) and how many barcodes (green dashes, right axis). **d)** A heatmap of how many barcodes were characterised for each mutation. **e)** A histogram of mutants by how many barcodes they had. **f)** Statistics on the completeness of the library. Overall we had >99% of the mutations in our lookup table.

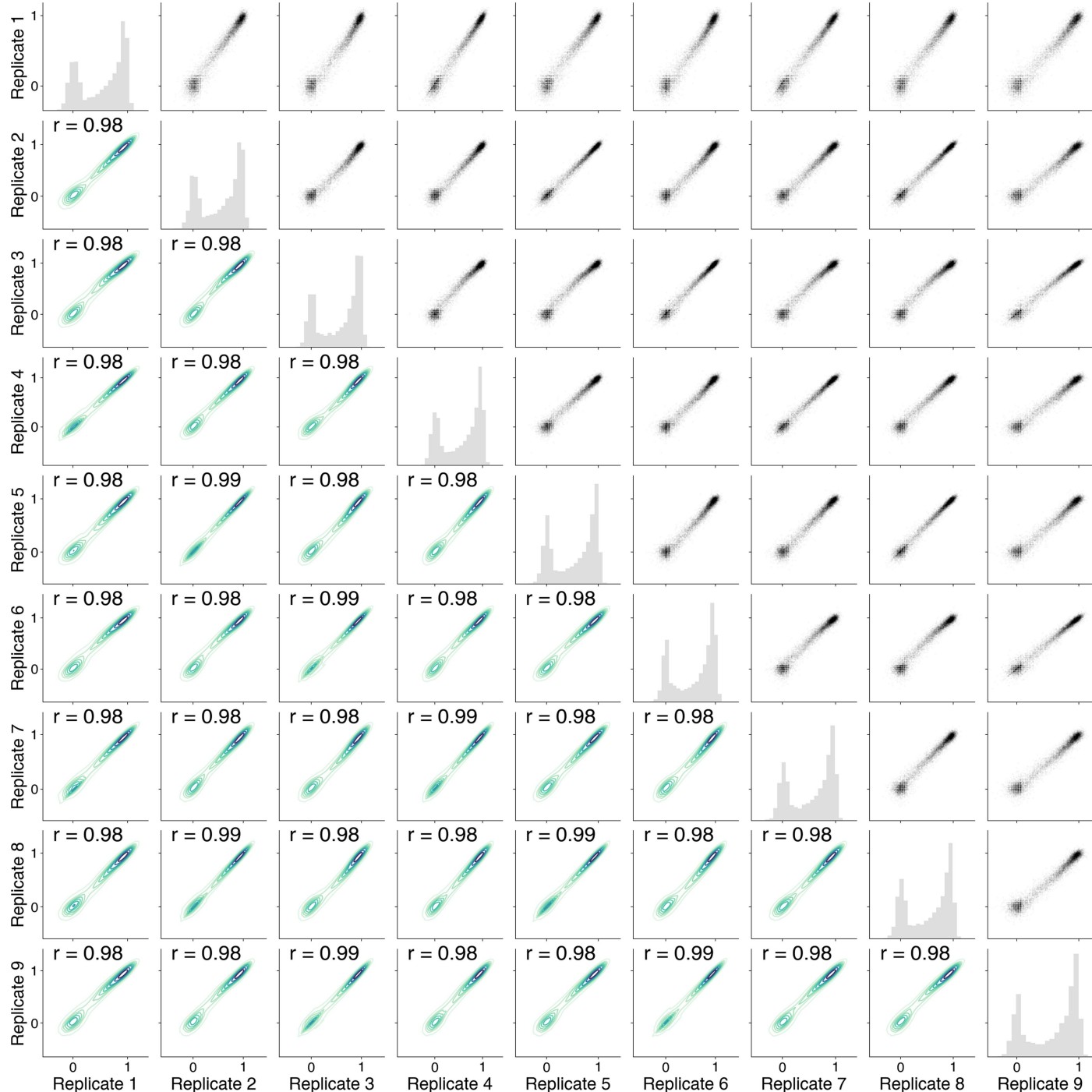

**Extended Data Fig. 5 | Pairplots of replicate fitness values.** Fitness values for each mutant are calculated as described in the methods for each replicate individually. These replicates are 3 sets of technical replicates of 3 biological replicates. Replicates 1, 4 and 7 are technical replicates (same with 2/5/8 and 3/6/9). Replicates 7–9 were collected on a different day. Pearson correlations reported for each pair of replicates. The distribution of fitness values is reported along the diagonal and pairwise correlations are reported between replicated off the diagonal. Pearson R is reported in the bottom-left half.

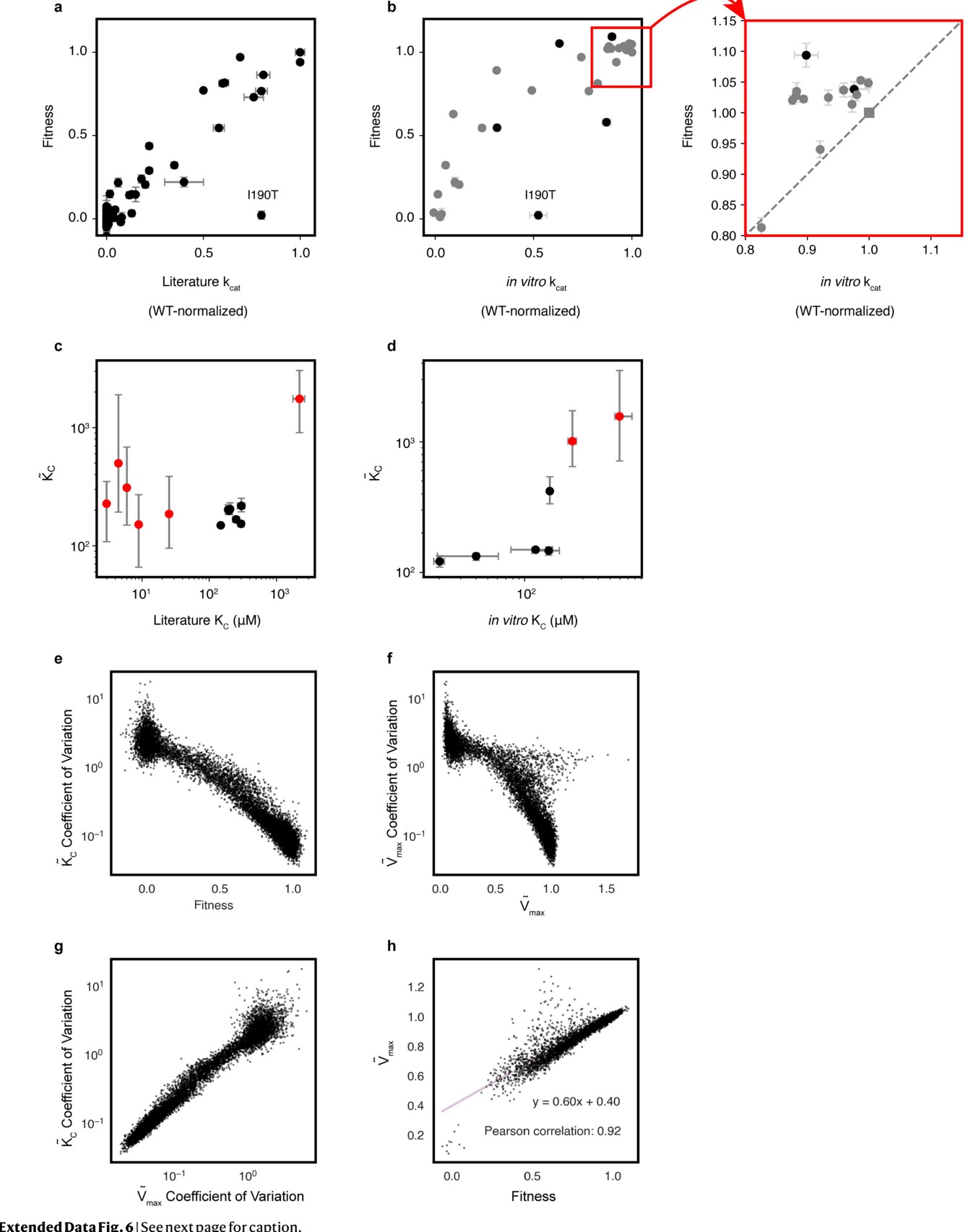

**Extended Data Fig. 6** | See next page for caption.

**Extended Data Fig. 6 | Comparisons between biochemically measured rubisco kinetic parameters and those same parameters as inferred from fitness values. a** and **b)** Fitness vs. $k_{cat}$ values, fitness error is the standard error of the mean for 9 replicates, **c** and **d)** $\widetilde{K}_C$ vs. $K_C$ values, $\widetilde{K}_C$ error bars reflect the inner quartiles of the bootstrap fits (see Methods). Measurements are from the literature in **a** and **c**, values are measured in this study by the spectrophotometric assay in **b** and **d**. Black points in **b** were purified 3 independent times (x-axis error bars are standard error), all other data in grey are from individual purifications and have no errors reported. Inset shows mutants with fitness values near or above 1 (WT-level). Dashed line indicates a 1:1 correspondence between fitness and in vitro measurements, WT is indicated with a square. X-axis error bars in **a** and **c** are taken from the literature when available. X-axis errors in **d** and Y-axis errors in **a-d** are explained in the methods. N = 3 biological replicates in all cases. Outlier mutation is labelled in **a** and **b** and is discussed in Methods. Red indicates $\widetilde{K}_C$ estimates with coefficient of variation >1. **e)** $\widetilde{K}_C$ coefficient of variation as a function of fitness. **f)** $\widetilde{V}_{max}$ coefficient of variation as a function of $\widetilde{V}_{max}$. **g)** $\widetilde{K}_C$ coefficient of variation as a function of fitness $\widetilde{V}_{max}$ coefficient of variation. **h)** Correlation of $\widetilde{V}_{max}$ and Fitness. Only mutants with a coefficient of variation <1 are plotted here; mutants with coefficients of variation >1 typically have low fitness and are thus harder to fit to a Michaelis-Menten model.

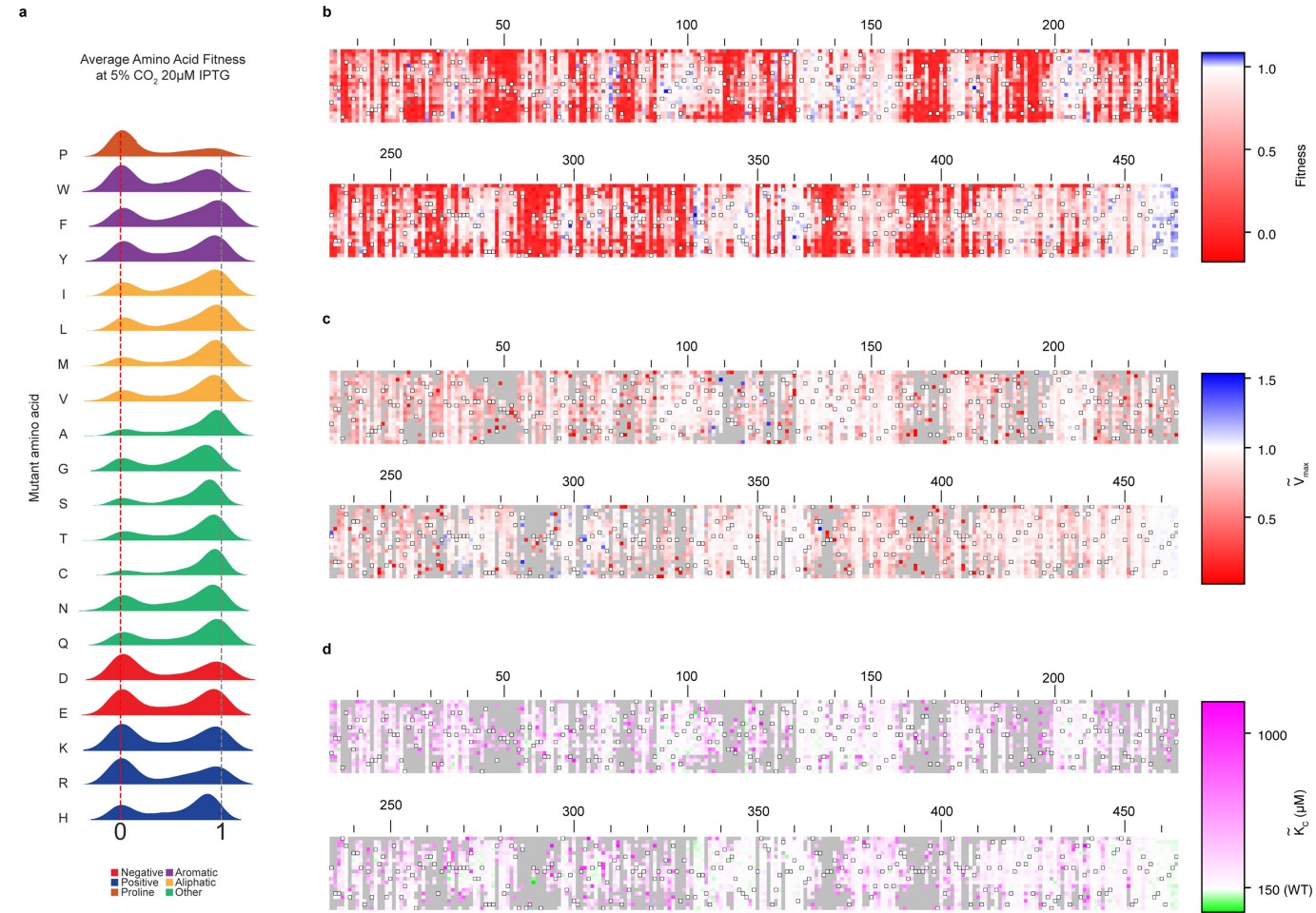

**Extended Data Fig. 7 | Histograms of fitness effects of mutations to each amino acid individually. a)** A histogram of fitness effects of all mutations to the specified amino acid (i.e. the plot for proline is the histogram of the fitness effects of mutations to proline at each position where there isn't a proline naturally). Plots are coloured by the biophysical properties of the amino acids. **b)** A heatmap of all fitness values. Fitness is the normalized enrichment value for selections carried out at 5% $CO_2$ with 20 μM IPTG. **c)** A heatmap of all $\widetilde{V}_{max}$ values. **d)** A heatmap of $\log(\widetilde{K}_C)$ values. $\widetilde{K}_C$ has units of μM $CO_2$.

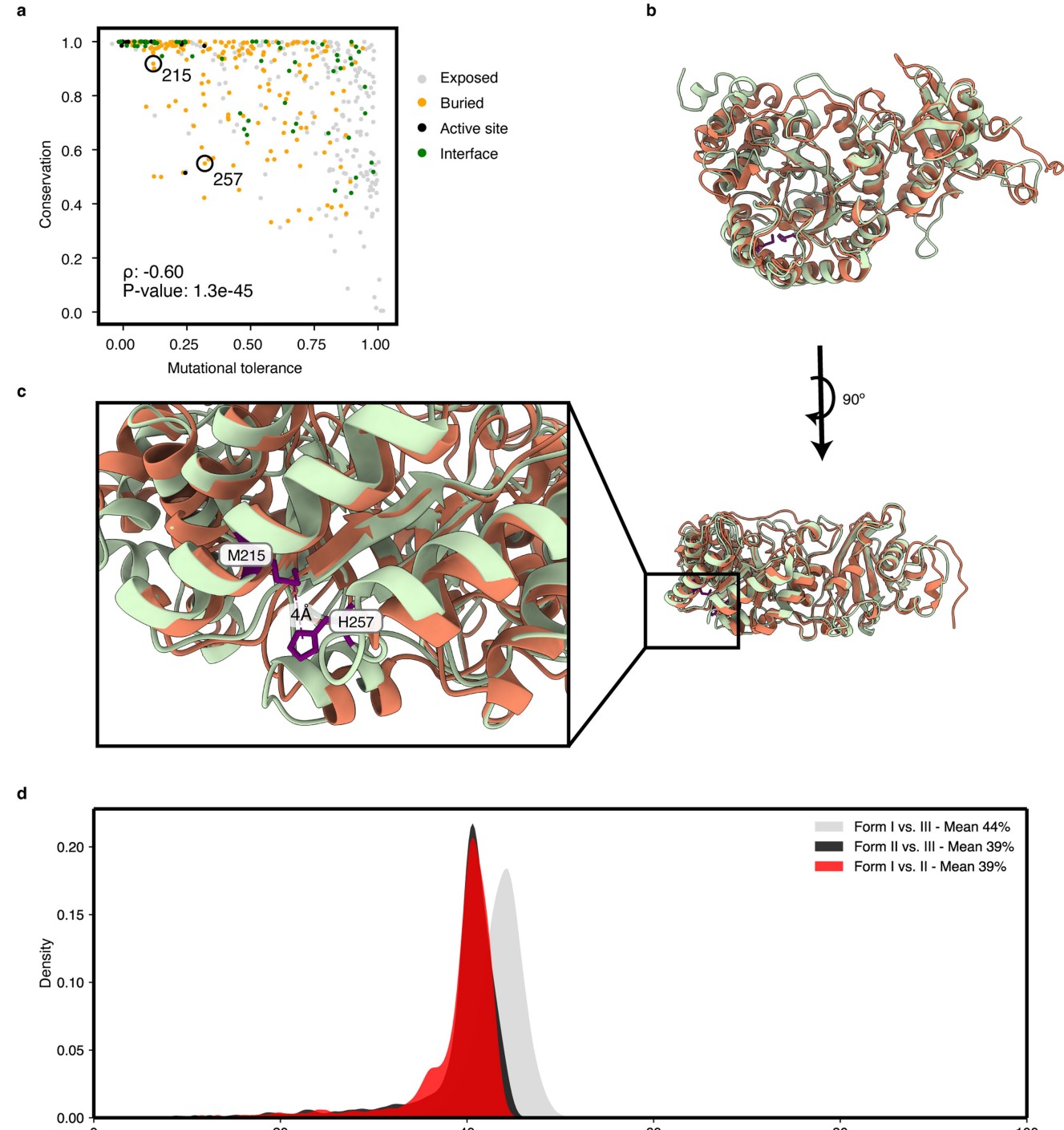

**Extended Data Fig. 8 | "Recent" evolution of a tertiary contact and phylogenetic comparisons. a)** Conservation vs. Tolerance among bacterial Form II rubiscos. As in Fig. 2c, mutational tolerance is the average fitness effect of all mutations at a given position. Here conservation is determined from an MSA of all Form II bacterial rubiscos (see methods). P-value is determined from the Spearman correlation and is thus a two-sided test. Positions 215 and 257 form a tertiary contact in *R. rubrum* and other Form II rubiscos and are thus more conserved than among all rubiscos. **b)** Alignment of 9RUB and 8RUC, *R. rubrum* (green) and spinach (orange) rubisco respectively. **c)** Rotated view and zoom of M215 and H257 from *R. rubrum*. The loop containing them in *R. rubrum* is truncated in spinach. **d)** Pairwise identities between rubisco sequences across Forms. Representative rubisco sequences from[5] were compared for pairwise identity. Form I sequences were picked to have a maximum sequence identity between one another of 85% in order to sample sequences more evenly (out of fear of oversampling plant sequences). Form II and III sequences were chosen randomly.

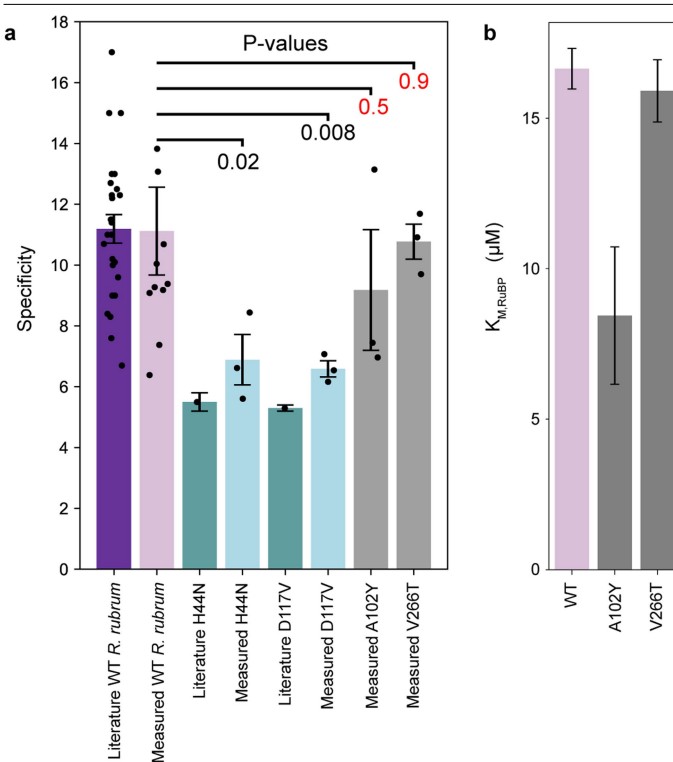

**Extended Data Fig. 9 | Specificity and $K_{M,RuBP}$ measurements for A102Y and V266T. a)** Specificity values measured by Membrane Inlet Mass Spectrometry (N = 3 for each mutant measured in this study). Comparisons to literature values are displayed when available. Literature data for WT is from[39]. Error bars represent the SEM of all measurements compiled in that published analysis. Literature data for H44N and D117V is from[24]. Error is taken from Extended Data Table 2 in that publication. P-values reflect a Welch's two-sided t-test in comparison to WT, with a permutation test to determine P-values. Red numbers indicate P > 0.05. **b)** $K_{M,RuBP}$ values fit from spectrophotometric assays of rubisco carboxylation along an 8 point RuBP titration. Each point in the titration was measured in technical triplicate. Error bars indicate the square root of the diagonals of the covariance matrix during fitting. All three triplicate measurements were used to perform the fit.

**Extended Data Table 1 | Assay mix composition**

| Component | Assay concentration | Source |
|---|---|---|
| EPPS buffer pH 8.0 | 100 mM | Alfa Aesar (Cat # J61296) |
| MgCl2 | 20 mM | Sigma Aldrich (Cat # M2670-500G) |
| Dithiothreitol | 0.5 mM | Bio Basic Canada inc. (Cat # DB0058) |
| ATP | 2 mM | Sigma Aldrich (Cat # A3377-5G) |
| Phosphocreatine | 10 mM | Sigma Aldrich (Cat # 27920-5G) |
| NADH | 1.7 mM | Merck (Cat # 481913-1GM) |
| Carbonic anhydrase | 0.1 mg/mL | Sigma Aldrich (Cat # C3934-100MG) |
| Creatine phosphokinase | 20 U/mL | Sigma Aldrich (Cat # C3755-35KU) |
| Glyceraldehyde 3-phosphate dehydrogenase | 20 U/mL | Sigma Aldrich (Cat # G2267-10KU) |
| 3-Phosphoglyceric phosphokinase | 20 U/mL | Sigma Aldrich (Cat # P7634-5KU) |

**Extended Data Table 2 | Enzyme kinetic parameters**

| | $k_{cat}$ (s$^{-1}$) | $K_C$ (µM CO$_2$) | $S_{C/O}$ | $K_{M,RuBP}$ (µM RuBP) |
|---|---|---|---|---|
| WT | 9.9 ± 0.4 | 148 ± 10 | 11 ± 1 | 16.6 ± 0.7 |
| V266T | 5.1 ± 0.1 | 87 ± 5 | 10.8 ± 0.6 | 16 ± 1 |
| A102Y | 1.71 ± 0.04 | 53 ± 3 | 9.2 ± 2 | 8 ± 2 |

# Reporting Summary

## Statistics

For all statistical analyses, confirm that the following items are present in the figure legend, table legend, main text, or Methods section.

| n/a | Confirmed | |
|---|---|---|
| ☐ | ☒ | The exact sample size (*n*) for each experimental group/condition, given as a discrete number and unit of measurement |
| ☐ | ☒ | A statement on whether measurements were taken from distinct samples or whether the same sample was measured repeatedly |
| ☐ | ☒ | The statistical test(s) used AND whether they are one- or two-sided *Only common tests should be described solely by name; describe more complex techniques in the Methods section.* |
| ☐ | ☒ | A description of all covariates tested |
| ☐ | ☒ | A description of any assumptions or corrections, such as tests of normality and adjustment for multiple comparisons |
| ☐ | ☒ | A full description of the statistical parameters including central tendency (e.g. means) or other basic estimates (e.g. regression coefficient) AND variation (e.g. standard deviation) or associated estimates of uncertainty (e.g. confidence intervals) |
| ☐ | ☒ | For null hypothesis testing, the test statistic (e.g. *F*, *t*, *r*) with confidence intervals, effect sizes, degrees of freedom and *P* value noted *Give P values as exact values whenever suitable.* |
| ☒ | ☐ | For Bayesian analysis, information on the choice of priors and Markov chain Monte Carlo settings |
| ☒ | ☐ | For hierarchical and complex designs, identification of the appropriate level for tests and full reporting of outcomes |
| ☐ | ☒ | Estimates of effect sizes (e.g. Cohen's *d*, Pearson's *r*), indicating how they were calculated |

*Our web collection on statistics for biologists contains articles on many of the points above.*

## Software and code

Policy information about availability of computer code

| Data collection | HMMER 3.4 |
|---|---|
| Data analysis | All data analysis components can be found at https://github.com/SavageLab/reads_processing<br>Packages used:<br>pandas 2.2.1<br>matplotlib 3.8.4<br>Biopython 1.81<br>numpy 1.26.4<br>scipy 1.12.0<br>seaborn 0.12.2<br>sklearn 1.2.2<br>pysam 0.21.0<br>Samtools 1.21 |

For manuscripts utilizing custom algorithms or software that are central to the research but not yet described in published literature, software must be made available to editors and reviewers. We strongly encourage code deposition in a community repository (e.g. GitHub). See the Nature Portfolio guidelines for submitting code & software for further information.

## Data

Policy information about availability of data

All manuscripts must include a data availability statement. This statement should provide the following information, where applicable:

- Accession codes, unique identifiers, or web links for publicly available datasets
- A description of any restrictions on data availability
- For clinical datasets or third party data, please ensure that the statement adheres to our policy

Sequences for our form II rubisco phylogeny were assembled from UniRef100
Our raw sequencing reads can be accessed on the NCBI SRA, accession ID: PRJNA1181558
All other data are available in the main text or the supplementary materials.

## Research involving human participants, their data, or biological material

Policy information about studies with human participants or human data. See also policy information about sex, gender (identity/presentation), and sexual orientation and race, ethnicity and racism.

| | |
|---|---|
| Reporting on sex and gender | N/A |
| Reporting on race, ethnicity, or other socially relevant groupings | N/A |
| Population characteristics | N/A |
| Recruitment | N/A |
| Ethics oversight | N/A |

Note that full information on the approval of the study protocol must also be provided in the manuscript.

# Field-specific reporting

Please select the one below that is the best fit for your research. If you are not sure, read the appropriate sections before making your selection.

☒ Life sciences   ☐ Behavioural & social sciences   ☐ Ecological, evolutionary & environmental sciences

For a reference copy of the document with all sections, see nature.com/documents/nr-reporting-summary-flat.pdf

# Life sciences study design

All studies must disclose on these points even when the disclosure is negative.

| | |
|---|---|
| Sample size | Triplicate or greater as indicated in figure panels. The only exception is the in vitro work in Extended Data figure 6B (grey points indicated in the figure are not repeated). Sample sizes were not chosen based on a calculation. Our default value for replication was triplicate. In the case of the experiment that generated the most critical dataset (figure 1G) we performed 9 replicates. |
| Data exclusions | No data was excluded, all data is available in the supplementary files. |
| Replication | All attempts at replication were successful. |
| Randomization | No randomization was used since it was not appropriate for this study, all analyses were done programmatically. |
| Blinding | Blinding was not relevant to our study, all analyses were done programmatically. |

# Reporting for specific materials, systems and methods

We require information from authors about some types of materials, experimental systems and methods used in many studies. Here, indicate whether each material, system or method listed is relevant to your study. If you are not sure if a list item applies to your research, read the appropriate section before selecting a response.

## Materials & experimental systems

| n/a | Involved in the study |
|---|---|
| ☐ | ☒ Antibodies |
| ☒ | ☐ Eukaryotic cell lines |
| ☒ | ☐ Palaeontology and archaeology |
| ☒ | ☐ Animals and other organisms |
| ☒ | ☐ Clinical data |
| ☒ | ☐ Dual use research of concern |
| ☒ | ☐ Plants |

## Methods

| n/a | Involved in the study |
|---|---|
| ☒ | ☐ ChIP-seq |
| ☒ | ☐ Flow cytometry |
| ☒ | ☐ MRI-based neuroimaging |

## Antibodies

| Antibodies used | Polyclonal Rabbit Anti-RbcL II, Agrisera, AS15 2955, Lot 2111<br>Polyclonal Goat to Rabbit IgG, abcam, ab205718, Lot GR3366929-1<br>Monoclonal clone 8E2/2, Mouse anti-DnaK, abcam, ab69617, 103741-3<br>Donkey anti-Mouse IgG-HRP, Santa Cruz BioTechnology, sc-2314, Lot C2012 |
|---|---|
| Validation | The anti-rbcL II antibody was validated against Alexandrium catenella, Amphidinium carterae, Chaetoceros neogracilis, Rhodobacter capsulatus, Rhodospirillum rubrum (relevant to this study)<br>Cho et al. (2021). SxtA localizes to chloroplasts and changes to its 3'UTR may reduce toxin biosynthesis in non-toxic Alexandrium catenella (Group I). Harmful Algae, 2021,101972,ISSN 1568-9883, https://doi.org/10.1016/j.hal.2020.101972. Immunolocalization<br>Bausch et al. (2019). Combined effects of simulated acidification and hypoxia on the harmful dinoflagellate Amphidinium carterae. Marine Biology, June 2019, 166:80.<br>Long et al. (2018). Carboxysome encapsulation of the CO2-fixing enzyme Rubisco in tobacco chloroplasts. Nat Commun. 2018 Sep 3;9(1):3570. doi: 10.1038/s41467-018-06044-0.<br><br>The anti-DnaK antibody has been used in 37 citations. The manufacturer states:<br>Mouse Monoclonal DNAK antibody. Suitable for WB and reacts with Recombinant full length protein - Escherichia coli, Escherichia coli samples. Cited in 37 publications. Immunogen corresponding to Full Length Protein corresponding to Escherichia coli K-12 dnaK. The antibody has been validated against E. coli DnaK which is relevant to this study. |

## Plants

| Seed stocks | N/A |
|---|---|
| Novel plant genotypes | N/A |
| Authentication | N/A |

