## [Peer Review File · Nature]

A map of the rubisco biochemical landscape

Corresponding Author: Professor David Savage

Parts of this Peer Review File have been redacted as indicated to maintain the confidentiality of unpublished data. This file contains all reviewer reports in order by version, followed by all author rebuttals in order by version.

Version 0:

Reviewer comments:

Referee #1

(Remarks to the Author)

“A map of the rubisco biochemical landscape” by Prywes et al. represents a breakthrough in analysis of the relationship between rubisco structure function. The authors report a high-throughput screen where rubisco activity is coupled to growth of *E. coli* and the performance of variants inferred from the abundance of barcodes via sequencing. The assay was used to analyze the impact of >99% of single amino acid substitutions of the Form II rubisco from *R. rubrum* on the affinity for CO₂ (K_c) in a single experiment. The authors use this to identify several point mutations which significantly improve K_c which were validated by in vitro assays, and provide a map of sites which are, and are not, amenable to mutation.

In my opinion, the novel approach used in this highly impressive manuscript is a leap forward in analysis of rubisco. Rubisco is the most abundant enzyme on the planet, and a key target for improvement in the field of photosynthesis. Researchers have tried to improve rubisco activity for decades, and several directed evolution systems have been developed. While there have been reports of mutations that increase rubisco catalytic activity (k_{cat}) and specificity (Wilson et al. 2016; 2018), most mutations identified resulted in improved folding and protein abundance. A key limitation of previous attempts is that they have been relatively low throughput. This assay moves beyond these attempts to achieve something previously impossible – measuring the impact of deep scanning mutagenesis on one of the catalytic parameters of rubisco. This has the potential to significantly improve our understanding of the catalytic trade-offs of rubisco (which are still debated), and there may be scope for investigating plant rubisco sequences which is important for bioengineering projects – with the caveat that tests would be required to determine if the cellular environment is sufficiently similar to the chloroplast stroma.

Overall, I thought the manuscript was excellent and highly exciting. The work was completed to a high standard. The text is well written, figures are clear, with the conclusions largely supported by the data. I only have a few comments.

An important consideration, which is briefly mentioned by the authors, is that the V_{max} (and their estimate) is dependent on enzyme activity and concentration. As with previous approaches, the assay reported here is therefore unable to distinguish between mutations which impact protein stability vs enzyme activity. Initial screens appear to have been done under saturating conditions to avoid this problem. This is only a minor drawback, given the data at least allow for an understanding of which mutations have a negative impact on performance, even if the precise cause is unclear. However, the limitation could perhaps be further clarified in the conclusion, as the initial mention of enzyme velocity and CO₂ affinity in the abstract could lead readers to assume that K_{cat} rather than V_{max} (a composite of activity and abundance) can be determined using the assay.

A second important consideration for bioengineering is the specificity of rubisco for CO₂ vs O₂. It is conceivable that rubisco is also catalyzing oxygenation in the assay, especially at lower CO₂ percentages during the early stages of growth (before dissolved O₂ declines in the flask). If so, what is the fate of 2-phosphoglycolate? Some of the authors previously discussed the possibility of a 2-PG salvage pathway in a similar *E. coli* mutant (Flamholz et al. 2020) and concluded it does not make a significant positive contribution to growth. This is encouraging, however it wasn't determined whether oxygenation negatively impacts fitness, which is important as it could have implications for estimates of K_c. Questions remain as to whether flux to 2-PG in the assay is insignificant, or has no impact on growth, and can be ignored, or if it could explain the discrepancies between estimated and measured K_c (Line 165, Fig 3G)? Testing this is complicated by the fact that *E. coli* is using oxygen for growth, however, some indication may be provided by performing an assay similar to Flamholz et al., along the lines of comparing growth of a rubisco containing strain on M9 medium + NO₃ with 0.5% CO₂/air, 0.5% CO₂/2% O₂, 0.5% CO₂/N₂. This could help provide confidence that oxygenation can be safely ignored.

Line 50-51 is true, but slightly misleading as it looks like the citation is supporting the inability to select for improvements. "Occasionally" would perhaps be a more accurate phrase than "rarely" and the authors may consider adding Wilson et al. 2018.

Fig 1F – Growth rate is dependent on V_{max} , and the conditions used (20 μ M IPTG 5% CO₂) appear to be saturating (extended data 2). It is therefore unclear how any mutants could have been identified as 'more fit' than WT under these conditions? are these instead within the range of biological noise?

Line 198: Related to the above comment, extended data 2 and 3 suggest that large increases in protein expression (perhaps 2-fold) result in no change in fitness. It is therefore unclear to me how the assay can identify variants with greater V_{max} values, or how these could be explained by increased protein abundance. Maybe I am missing something, but it would help to clarify.

Line 556: typo? "Error bars in C determined as"

Line 667: typo? Unless it was intended to mean only those data with a poor fit are plotted?

References:

Wilson et al. 2016 10.1038/srep22284

Wilson et al. 2018 10.1074/jbc.M117.810861

Flamholz et al. 2020 <https://doi.org/10.7554/eLife.59882>

Referee #2

(Remarks to the Author)

Summary

This study couples *E. coli* growth to Rubisco function, and then performs in vivo saturating mutagenesis of *Rhodospirillum rubrum* Rubisco. This permits the use of *E. coli* fitness as a proxy for V_{cmax} and K_c . V_{cmax} correlated well with fitness. However, from solely a fitness measurement, there is inherent uncertainty as to whether this will be driven by (and to what extent) k_{cat} or Rubisco solubility. The relative location and resultant fitness of these substitutions are discussed in the context of amino acid sequence and 3D protein structure. This work is exciting for a number of reasons including indicating that Rubisco's functional landscape can definitely be further explored (and in this study different function was obtained from only single point substitutions), and that it provides a relatively high-throughput method for estimating Rubisco kinetics in vivo that could be extended to explore the mutational landscape of vascular plant Rubiscos. This work is of exceptional quality and importance. I thank the authors for such careful manuscript preparation. The work is excellently communicated and display items are well-presented.

I provide a list of very minor revisions / suggestions:

1. Line 48. Suggest writing "...towards vascular plant Rubiscos."
2. Line 69 or so. Is it worthwhile mentioning that you chose you chose an IPTG concentration for expression (20 μ M) where getting high Rubisco expression, but also seems to be in linear response part of the expression curve? (if indeed that was your rationale for this choice). It seemed like a good choice to get maximum fitness sensitivity from your screen.
3. Line 86. Do you mean 5A?
4. Line 130. "Extended Data Fig. 1 right panel" (there is no C)
5. Figure 2A. Consider marking active sites, e.g. with a small triangle / asterisk, unless this obscures too much of the average fitness data.
6. Figure 2C. Consider a darker green for closed loop 6, as poor contrast.
7. Line 163. Consider referencing Fig. 3F (as well as Fig.3 E).
8. Line 163. Specify how many mutants for which the K_c values were much lower than expected. If there are more than A102Y and V226T, indicate these in Figure 3F.
9. Figure 3B. Consider more distinct color scale. I cannot tell the difference from 0.2-1%.
10. Figure 3G. Please indicate location of residue 102.
11. Line 237. Provide brief detail about how Δr_{pi} was made. Text currently sounds like it was made in another study.
12. Line 285. How was CO₂ supplied to the cultures?
13. Line 529. I cannot see the symmetry labelled in this figure.

14. Line 539. State what error bars represent (as you have done in Extended Fig. 4 legend).
15. Line 545. Why does DnaK also increase with increasing IPTG?
16. Line 550. Consider extending the color scale, I cannot see the difference between most of these colors.
17. Line 556. Second description of error bars is D (not C).
18. Line 627. "Plots are colored by amino acid property" (or similar)
19. Line 667. Do you mean: "those mutants with a coefficient >1 typically..."

General comments

Which kinetic assays did you use for which data? Both spec and C14 methods are listed in the methods section, but these generally give slightly different values for the same enzyme. Further, the C14 method describes deriving K_c , whereas the spec assay is deriving K_c air. Is the in vivo fitness essentially estimating K_c air, rather than K_c ? Regardless, this is unlikely to account for the large difference in predicted and measured values for V266.

The only additional data that would strengthen this (and I hesitate to ask for it, so I include as a suggestion, not requirement), is expression data for the 7 mutants experimentally validated in this study. This would confirm that the V_{max} inferred from fitness is indeed driven by increased solubility, given the reduced k_{cat} .

Do you have any suggestions for why K_c may have been so different for some of your 7 mutants?

I am curious as to how oxygenation kinetics could be captured in this, or a derivative screen.

Referee #3

(Remarks to the Author)

RuBisCO, one of the most abundant enzymes on Earth, plays a key role in global carbon fixation. Nevertheless, it is characterized by slow catalytic turnover and poor CO₂ affinity. Increasing its activity (and minimizing an undesired side reaction with molecular oxygen) could potentially improve photosynthetic efficiency and hence crop yields, but engineering RuBisCO has proved challenging. In this study, Prywes et al. have addressed this issue by applying genetic selection methods to map sequence-function relationships in a bacterial RuBisCO in high throughput. Using an engineered *E. coli* as a host, they successfully assayed >99% of all single amino acid substitutions in the enzyme, determining relative velocities and CO₂ affinity parameters for thousands of variants in parallel. While no enzymes with higher specific activities were found, a few substitutions were identified that conferred higher CO₂ affinity and maintained WT-like fitness levels. This system thus provides a powerful platform for high-throughput engineering and systematic exploration of the tradeoffs that constrain RuBisCO's evolution. The work appears to have been carried out with care and admirable attention to detail. If the concerns outlined below can be successfully addressed, the manuscript may therefore be suitable for publication in *Nature*.

Critique:

- 1) The acronyms Ri5P, Ru5P and RuBP in Fig.1 should be defined somewhere in the legend or the text.
- 2) Fig. 1 legend: k_{cat} is a rate constant, not a "catalytic rate".
- 3) Fig.2 legend: panels C) and D) are mislabeled.
- 4) Lines 172-175: If no binding site exists for CO₂, how can one speak of "improved affinity"? Why would the kinetic plots show saturation if CO₂ does not bind to the enzyme? In such a case, wouldn't one expect a linear increase in rate with increasing CO₂ concentration?
- 5) Fig. 3B: The population of active variants appears to shift to the left somewhat as the CO₂ concentration increases, suggesting that it becomes less fit. Is this expected? Is bacterial RuBisCO subject to autoinhibition under these conditions?
- 6) Fig. 3D and E: Does 1 mM RBP, the concentration used for the in vitro kinetic measurements, saturate RuBisCO? How does this concentration compare to the concentration of RBP in vivo? Could differences in the in vitro and in vivo RBP concentrations explain the "unexpectedly" lower K_c values measured in vitro (line 163)?
- 7) Line 190: why "also"? The three variants mentioned in the previous sentence show an apparent increase (better than "improvement") in CO₂ affinity, not a decrease.
- 8) Line 199: "Our K_c predictions were isolated from expression effects because mutants were judged individually by their relative performance across a CO₂ titration and were thus more accurate." This seems like an odd explanation. Isn't K_c , by definition, independent of enzyme concentration and therefore expression effects?

9) Lines 200-203: Because k_{cat} for the characterized variants decreased more than CO_2 affinity increased, catalytic efficiency, defined as k_{cat}/K_m , dropped (Extended Data Table 2). This finding argues against the statement in the abstract that “non-trivial kinetic improvements are readily accessible.” The mutations do not constitute a kinetic improvement; as noted by the authors, high flux was likely maintained by increasing protein expression, which is much easier to accomplish than improving catalytic efficiency. Instead, the results appear to underscore the difficulty of engineering more efficient RuBisCOs, at least through single amino acid substitutions, and speak to the need to screen more complex libraries containing combinations of mutations to find variants with improved catalytic efficiency.

10) How do the mutations that increase CO_2 efficiency affect the competing oxygenation reaction? Given the importance of suppressing this undesired side reaction, the authors should measure the CO_2 vs O_2 specificity.

(Remarks on code availability)

Unclear what “code” I was supposed to review. If you mean the Reporting Summary, I looked at it but I do not find this information useful.

Version 1:

Reviewer comments:

Referee #1

(Remarks to the Author)

The authors have added a substantial amount of additional supporting data and successfully address my comments. In particular:

- Data showing assay also picks up mutations in protein expression/stability and inclusion in manuscript and text (extended data 2).
- Show that the plateau in growth rate at high IPTG concentrations is likely due to protein burden/toxicity rather than a maximum limit on enzyme activity, as indicated by the relationship between yield and expression, and growth rate and CO_2 concentration (extended data 2).
- O_2 titrations which show that 2-PG production is likely to have negative consequences on growth and inclusion of this data plus limitation (extended data 2).

I congratulate the authors on an outstanding piece of work with great potential to further our understanding of the relationship between rubisco protein sequence and properties.

(Remarks on code availability)

n/a

Referee #2

(Remarks to the Author)

All points for reviewer # 2 were more than satisfactorily addressed by the authors.

Referee #3

(Remarks to the Author)

The authors have largely addressed my concerns. The one remaining issue regards the purported absence of a CO_2 binding site in RuBisCo. If true, equating the K_c parameter with CO_2 “affinity”, as is done consistently throughout the manuscript, should be avoided. This is an unnecessary and misleading simplification. As clearly explained in the rebuttal, there is no pre-equilibrium of bound CO_2 , so K_c -- formally the CO_2 concentration at which the reaction velocity is half maximal -- is not the dissociation constant for CO_2 . Instead, this parameter is determined by the CO_2 on-rate, the subsequent rate of its reaction with activated RuBP, and presumably the binding constant for RuBP.

Point by point responses for Prywes et al. 2024

Original reviewer comments are in black

Authors' responses and updates are in red

Changes to the manuscript text are noted in blue

Updated figures and tables:

Extended Data Figure 2: Yield data added

Extended Data Figure 2: I164T data added

Extended Data Figure 2: [O₂] titration growth curve added

Extended Data Figure 6: Inset added

Extended Data Figure 9 added: Specificity and $K_{M,RuBP}$ data

Extended Data table 2 placed into Extended Data Fig. 9: Specificity values added

File S1 was updated to fill in an error (the VLOOKUP from google sheets did not work when converted to excel).

We combined several Extended Data Figures (2+3+4), (13+18), (table 2+17+ $K_{M,RuBP}$ data), (8+16), (9+10+11+12), and removed 14,15

Referee #1

“A map of the rubisco biochemical landscape” by Prywes et al. represents a breakthrough in analysis of the relationship between rubisco structure function. The authors report a high-throughput screen where rubisco activity is coupled to growth of *E. coli* and the performance of variants inferred from the abundance of barcodes via sequencing. The assay was used to analyze the impact of >99% of single amino acid substitutions of the Form II rubisco from *R. rubrum* on the affinity for CO₂ (K_c) in a single experiment. The authors use this to identify several point mutations which significantly improve K_c which were validated by in vitro assays, and provide a map of sites which are, and are not, amenable to mutation.

In my opinion, the novel approach used in this highly impressive manuscript is a leap forward in analysis of rubisco. Rubisco is the most abundant enzyme on the planet, and a key target for improvement in the field of photosynthesis. Researchers have tried to improve rubisco activity for decades, and several directed evolution systems have been developed. While there have been reports of mutations that increase rubisco catalytic activity (k_{cat}) and specificity (Wilson et al. 2016; 2018), most mutations identified resulted in improved folding and protein abundance. A key limitation of previous attempts is that they have been relatively low throughput. This assay moves beyond these attempts to achieve something previously impossible – measuring the impact of deep scanning mutagenesis on one of the catalytic parameters of rubisco. This has the potential to significantly improve our understanding of the catalytic trade-offs of rubisco (which are still debated), and there may be scope for investigating plant rubisco sequences which is important for bioengineering projects – with the caveat that tests would be required to determine if the cellular environment is sufficiently similar to the chloroplast stroma.

Overall, I thought the manuscript was excellent and highly exciting. The work was completed to a high standard. The text is well written, figures are clear, with the conclusions largely supported by the data. I only have a few comments.

An important consideration, which is briefly mentioned by the authors, is that the V_{max} (and their estimate) is dependent on enzyme activity and concentration. As with previous approaches, the assay reported here is therefore unable to distinguish between mutations which impact protein stability vs enzyme activity. Initial screens appear to have been done under saturating conditions to avoid this problem. This is only a minor drawback, given the data at least allow for an understanding of which mutations have a negative impact on performance, even if the precise cause is unclear. However, the limitation could perhaps be further clarified in the conclusion, as the initial mention of enzyme velocity and CO₂ affinity in the abstract could lead readers to assume that K_{cat} rather than V_{max} (a composite of activity and abundance) can be determined using the assay.

We thank the reviewer for their attention to this key detail. Although we did observe a general correspondence between k_{cat} and fitness indicating that expression levels are generally constant (at least in the condition where we measured fitness, 20μM IPTG), we agree that this consideration merits further discussion beyond the current lines 169-171 and 200-201. We have thus added a paragraph (line 225) in the conclusion to address the issue and also updated Extended Data Fig. 2 with new results. These additions are collected in Reviewer Figure 1 for viewing convenience below.

Specifically, we were able to identify examples of mutations that had an effect on the expression profile. Notably, I164T caused improved growth with increasing [IPTG], even at very high levels where WT growth suffers (Reviewer Figure 1A). I164T has a k_{cat} measured in the literature at ~6% of WT (Chène et al. 1997), so the fact that their growth rates converged seemed to indicate that the intracellular concentration of active I164T enzyme was much higher than that of WT at high [IPTG]. We quantified the expression of I164T and found that, indeed, it outpaced WT with

increasing [IPTG] (Reviewer Figure 1B,C). It remains to be determined how common this kind of mutant effect is. Other studies measuring mutational effects have found many examples of effects on expression (Faure et al. 2023), but it is likely very protein-dependent.

Reviewer figure 1: Effects of rubisco expression level on the growth of ΔRPI . **A)** I164T has higher growth rates with higher [IPTG] while WT declines at high induction. **B)** This western blot indicates that I164T expresses at much higher levels than WT and explains its fitness values that seem to contradict its low reported k_{cat} (6% of the WT value). **C)** Quantification of the expression of rbcL in **B** normalized to DnaK. **D)** Growth curves of ΔRPI at different IPTG-induction levels. This is the underlying data for extended figure 2C. **E)** While increasing IPTG levels do not reduce the growth rate they do reduce growth yields and cause a negative growth phenotype.

A second important consideration for bioengineering is the specificity of rubisco for CO₂ vs O₂. It is conceivable that rubisco is also catalyzing oxygenation in the assay, especially at lower CO₂ percentages during the early stages of growth (before dissolved O₂ declines in the flask). If so, what is the fate of 2-phosphoglycolate? Some of the authors previously discussed the possibility of a 2-PG salvage pathway in a similar *E. coli* mutant (Flamholz et al. 2020) and concluded it does not make a significant positive contribution to growth. This is encouraging, however it wasn't determined whether oxygenation negatively impacts fitness, which is important as it could have implications for estimates of K_c . Questions remain as to whether flux to 2-PG in the assay is insignificant, or has no impact on growth, and can be ignored, or if it could explain the discrepancies between estimated and measured K_c (Line 165, Fig 3G)? Testing this is complicated by the fact that *E. coli* is using oxygen for growth, however, some indication may be provided by performing an assay similar to Flamholz et al., along the lines of comparing growth of a rubisco containing strain on M9 medium + NO₃ with 0.5% CO₂/air, 0.5% CO₂/2% O₂, 0.5% CO₂/N₂. This could help provide confidence that oxygenation can be safely ignored.

Like expression level, the effect of oxygenation is a key consideration for the use of *E. coli* as a tool for rubisco biochemistry. We thank the reviewers for the assay recommendation and have performed a series of growth curves as suggested (we include this titration as Extended Data Figure 2E in the revised manuscript). We titrated O₂ from 2% up to 20% and found that while the growth of a control strain (BW25113) expressing GFP had increasing yield up to 20% O₂, *Δrpi* expressing WT *R. rubrum* rubisco had improved growth rates and yields up to 10% O₂ followed by a decline. The increase in growth rate from 2-10% O₂ may be a result of improved cellular respiration. The subsequent decline could reflect a deleterious effect of 2-PG production from rubisco oxygenation, either in the form of an opportunity cost (2-PG is less valuable to *E. coli* than 3-PG) or of toxicity. Toxicity may be the result of specific interactions with other cellular components or simply as a metabolic imbalance resulting from 2-PG buildup. We have included a paragraph in the conclusion (line 236) to address this limitation in our study.

The question gets at the root of a potential ambiguity with the strain and resulting two mutants studied in detail in our work (A102Y, V266T). To resolve this issue, we also carried out a biochemical analysis of S_{C/O} for these mutants (new Extended Data Figure 9), which shows that their specificities are the same as that of WT, despite the change in their carboxylation efficiencies ($k_{cat,C}/K_C$). These data can be found below, where asked for by Reviewer 3.

Finally, we have also begun working on an O₂ titration experiment and present some very preliminary data from those efforts in an answer to Reviewer 2 below.

[REDACTED]

Line 50-51 is true, but slightly misleading as it looks like the citation is supporting the inability to select for improvements. “Occasionally” would perhaps be a more accurate phrase than “rarely” and the authors may consider adding Wilson et al. 2018.

We thank the reviewers for this suggestion and have updated the manuscript accordingly.

Fig 1F – Growth rate is dependent on V_{max}, and the conditions used (20 μM IPTG 5% CO₂) appear to be saturating (extended data 2). It is therefore unclear how any mutants could have been identified as ‘more fit’ than WT under these conditions? are these instead within the range of biological noise?

This question is a critical one for the future use of this strain to distinguish between mutants with higher fitness. Since we did not measure expression level in this study we were unable to distinguish between mutants with high fitness as a result of improved k_{cat} or improved expression (Reviewer Figure 1), but this question goes further to ask if fitness has plateaued in our study.

The mutants identified in this study to have higher V_{max} values than WT were very consistent across replicates and did not appear to be merely noise. In a significance analysis, one mutant had a higher fitness than WT, after Bonferroni correction for multiple hypothesis testing (Reviewer Figure 3A). More mutants were significantly more fit when using a Benjamini-Hochberg or Benjamini-Yekutieli correction with a 1% false discovery rate. However, followup analysis on a subset of these mutants revealed that they had the same or lower k_{cat} values than WT (Reviewer Figure 3B) leading us to conclude that their apparently improved fitness is a result of improved expression or reduced proteolytic toxicity. The data in Reviewer Figure 3B was presented in the original manuscript and referred to at line 201-203 but was perhaps missed. To better draw attention to this issue we have expanded the text at line 120 and have edited Extended Data Fig. 6B with an inset and diagonal line to highlight the experimental result.

In Figure 1E we observe that fitness increases as a function of *in vitro* k_{cat} without an obvious plateau. However, it is difficult to tell. In preliminary data outside of the scope of this study we

repeated the library selection at a titration of [IPTG]. These results are complicated by the counteracting effects of beneficial rubisco flux and detrimental rubisco overexpression. However, when we repeat the analysis of Figure 1E along an [IPTG] titration we do not observe an apparent plateau (Reviewer Figure 3C), indicating that the benefit of rubisco flux does not saturate. How successful the strain remains at identifying substantially higher k_{cat} s remains unknown, and we have added text to the conclusion section to clarify this point (lines 233-234).

Reviewer Figure 3: Detection of “improved” mutants and plateauing **A** A volcano plot showing the fitness effects and significances of mutants just above and below WT. Significance thresholds for the Bonferroni correction, Benjamini-Yekutieli and Benjamini-Hochberg (False discovery rate = 1%) are indicated. Positive mutants above those thresholds are indicated in red, blue and green respectively. **B** Zoom-in for extended figure 6B. All mutants identified as having high V_{max} values had k_{cat} values that were similar to or slightly below those of WT in an *in vitro* test. **C** The fitness of mutants with lower reported k_{cat} values increased progressively with increasing [IPTG]. However, there was no concentration of IPTG where fitness plateaued as a function of k_{cat} .

Line 198: Related to the above comment, extended data 2 and 3 suggest that large increases in protein expression (perhaps 2-fold) result in no change in fitness. It is therefore unclear to me how the assay can identify variants with greater V_{max} values, or how these could be explained by increased protein abundance. Maybe I am missing something, but it would help to clarify.

As the reviewers point out, there is an apparent plateau in growth rate as a function of rubisco expression. However, this result is misleading because the shape of the growth curves continues to change with increasing [IPTG] (Reviewer Figure 1D). As rubisco expression rises, growth rates plateau but growth yields begin to diminish (Reviewer Figure 1E). We suspect that this is a consequence of proteolytic stress from rubisco overexpression. Based on the result in Reviewer Figure 3C (see above) we suspect that there is no plateau but refrain from commenting in this manuscript.

We have updated Extended Data Fig. 2 to clarify and have updated the text (lines 78-79) accordingly.

Line 556: typo? “Error bars in C determined as”

We thank the reviewers for catching this error, edited to “L” in the updated version of Extended Data Fig. 2.

Line 667: typo? Unless it was intended to mean only those data with a poor fit are plotted?

The x-axes of Extended Data Figure 16 A,B were accidentally obscured, we have corrected that error in this resubmission in Extended Data Fig. 6.

Indeed we only considered higher quality K_c fits (those with lower coefficients of variation). We set an upper threshold of 1 for the coefficient of variation based on Jones et al. 2020. We report the coefficient of variation (CoV) based on the variability observed when recomputing the Michaelis-Menten parameters 1100 times with different pseudocounts and lower limit read thresholds across biological replicates (see Methods). In the supplementary data we include estimates for all mutants (even those with lower quality estimates) along with CoV.

Referee #2

Summary

This study couples *E. coli* growth to Rubisco function, and then performs in vivo saturating mutagenesis of *Rhodospirillum rubrum* Rubisco. This permits the use of *E. coli* fitness as a proxy for V_{cmax} and K_c . V_{cmax} correlated well with fitness. However, from solely a fitness measurement, there is inherent uncertainty as to whether this will be driven by (and to what extent) k_{cat} or Rubisco solubility. The relative location and resultant fitness of these substitutions are discussed in the context of amino acid sequence and 3D protein structure. This work is exciting for a number of reasons including indicating that Rubisco's functional landscape can definitely be further explored (and in this study different function was obtained from only single point substitutions), and that it provides a relatively high-throughput method for estimating Rubisco kinetics in vivo that could be extended to explore the mutational landscape of vascular plant Rubiscos. This work is of exceptional quality and importance. I thank the authors for such careful manuscript preparation. The work is excellently communicated and display items are well-presented.

I provide a list of very minor revisions / suggestions:

1. Line 48. Suggest writing "...towards vascular plant Rubiscos."

We thank the reviewers for the suggestion and have edited the manuscript accordingly.

2. Line 69 or so. Is it worthwhile mentioning that you chose you chose an IPTG concentration for expression (20 μM) where getting high Rubisco expression, but also seems to be in linear response part of the expression curve? (if indeed that was your rationale for this choice). It seemed like a good choice to get maximum fitness sensitivity from your screen.

The choice of 20 μM IPTG was indeed meant to be in the linear response regime. We were also trying to avoid expressing rubisco at too low of a level, where small differences in expression might have large effects, and too high of a level, where proteolytic stress confounds the results. Please see the responses to reviewer #1 for a more detailed explanation (along with Reviewer Figure 1).

We thank the reviewers for this clarification and have updated the text accordingly (lines 108-110).

3. Line 86. Do you mean 5A?

We did mean 5A and thank the reviewers for spotting the error.

4. Line 130. "Extended Data Fig. 1 right panel" (there is no C)

We thank the reviewers for the suggestion and have edited the manuscript accordingly.

5. Figure 2A. Consider marking active sites, e.g. with a small triangle / asterisk, unless this obscures too much of the average fitness data.

We thank the reviewers for the suggestion and have updated the figure accordingly.

6. Figure 2C. Consider a darker green for closed loop 6, as poor contrast.

We thank the reviewers for the suggestion and have updated the figure accordingly.

7. Line 163. Consider referencing Fig. 3F (as well as Fig.3 E).

We thank the reviewers for the suggestion and have edited the manuscript accordingly.

8. Line 163. Specify how many mutants for which the K_C values were much lower than expected. If there are more than A102Y and V226T, indicate these in Figure 3F.

We have updated figure 4A and the text (lines 167-168) to indicate that there were 4 mutants in our full library that had significantly lower K_C estimates. As far as mutants where we measured their K_C *in vitro* and found that it was substantially lower, there were A102Y and V266T. We speculate that A289C might similarly have a surprisingly low K_C *in vitro* but refrain from including that in the text.

9. Figure 3B. Consider more distinct color scale. I cannot tell the difference from 0.2-1%.

We thank the reviewers for the suggestion and have updated the figure accordingly.

10. Figure 3G. Please indicate location of residue 102.

We thank the reviewers for the suggestion and have updated the figure accordingly.

11. Line 237. Provide brief detail about how Δrpi was made. Text currently sounds like it was made in another study.

Δrpi was engineered by the Milo laboratory and was never specifically published on. However, a subsequent strain bearing additional modifications, CCMB1, was developed from *Δrpi* for (Flamholz et al. 2020). We have thus added a citation to the methods section to indicate this more clearly, along with additional details regarding strain construction (lines 259-262). Briefly, P1 transduction was used to remove *edd* and *rpiA* from the Keio collection strain already lacking *rpiB*. After each knockout the kanamycin marker was removed by curing with pCP20.

12. Line 285. How was CO₂ supplied to the cultures?

5mL cultures were grown in 11mL culture tubes at 37 °C in a Percival Intellus Incubator at different CO₂ concentrations on a New Brunswick Scientific Innova 2000 shaker at 250 RPM at an angle of 60°. Cultures were grown until they reached an OD₆₀₀ at 5 mL of 1.2 +/- 0.2.

We have amended the methods section to reflect this (lines 318-320).

13. Line 529. I cannot see the symmetry labelled in this figure.

We used a two-fold axis symbol: . The text of the caption has been edited for clarity.

14. Line 539. State what error bars represent (as you have done in Extended Fig. 4 legend).

Error here was SEM for four replicates, we have updated the caption accordingly.

15. Line 545. Why does DnaK also increase with increasing IPTG?

The samples were not normalized to the same OD and cultures grew faster with higher IPTG so more cells were lysed and loaded into the wells with more IPTG. We used DnaK as a loading control to account for this systematic effect.

16. Line 550. Consider extending the color scale, I cannot see the difference between most of these colors.

We thank the reviewers for the suggestion and have updated the figure accordingly.

17. Line 556. Second description of error bars is D (not C).

We thank the reviewer for spotting the error and have edited the manuscript accordingly.

18. Line 627. "Plots are colored by amino acid property" (or similar)

We have updated the text with this improvement.

19. Line 667. Do you mean: "those mutants with a coefficient >1 typically..."

We thank the reviewer for spotting the error and have edited the manuscript accordingly.

General comments

Which kinetic assays did you use for which data? Both spec and C14 methods are listed in the methods section, but these generally give slightly different values for the same enzyme. Further, the C14 method describes deriving K_c , whereas the spec assay is deriving $K_{c,air}$. Is the in vivo fitness essentially estimating $K_{c,air}$, rather than K_c ? Regardless, this is unlikely to account for the large difference in predicted and measured values for V266.

The spec assay was used for Figure 3F and Extended Figure 6B,D and 9B. The ^{14}C method was used for Figure 4C and extended data table 2. In the spec assay as well as the ^{14}C assay we flushed out O_2 with N_2 so the measurements are actual K_c measurements and not $K_{c,air}$. The

estimates from the CO₂ titration are $K_{C,air}$, this distinction may be contained in the ~ symbol above the K_C symbol. We have also amended the text to highlight this distinction on lines 185, 214, 438, 458, 673 and 710.

The only additional data that would strengthen this (and I hesitate to ask for it, so I include as a suggestion, not requirement), is expression data for the 7 mutants experimentally validated in this study. This would confirm that the V_{max} inferred from fitness is indeed driven by increased solubility, given the reduced k_{cat} .

We measured k_{cat} values for mutants with high V_{max} and fitnesses (Extended figure 6B and Reviewer Figure 3B). None of the “superior” mutants turned out to have higher k_{cat} values indicating that their highly significant (see Reviewer Figure 3A above) V_{max} or fitness improvements come from expression differences.

We agree with the reviewer that exploring the effect of a mutation in terms of either catalytic effect versus a more nebulous expression effect (stability, degradation rate, etc.) is an important topic. To this end, we have included new panels in Extended Data Fig. 2 where we identify a mutant that expresses to higher levels than WT (see also Reviewer Figure 1A-C and discussion in responses to reviewer #1). We have added a paragraph to the conclusion to address this concern (lines 225-235).

Do you have any suggestions for why K_C may have been so different for some of your 7 mutants?

We do not yet know why our Michaelis-Menten model underestimates the effect on K_C . It is possible that our assumptions about the relationship between rubisco flux and growth rate are oversimplified. It is also possible that the presence of oxygen creates a sort of systematic bias that we have not accounted for. Local production of CO₂ through respiration could have an effect, though we find that doubtful in the face of rapid diffusion. As reviewer 3 points out, we do not know the precise concentration of RuBP, if it is below the reported K_M (15 μ M), that could affect our estimates as well.

We are currently performing additional gas titrations in the hopes of finding clues that can explain this systematic effect. One thing that we can rule out is a strong specificity effect, A102Y and V266T have specificities that are very similar to WT (new Extended Data Figure 9). Nevertheless, the fact that the lowest predicted K_C 's flagged by the model are borne out by the biochemical measurements gives us confidence in the trends despite the systematic quantitative deviations.

If the reviewer is asking about the underlying mechanism by which the mutations (say at position 102 or 266) alter affinity, we are currently unsure. In the manuscript we speculate that their position near the C2 axis could affect monomer-monomer vibration which could allosterically affect the gas addition step either by changing the active site entry aperture or by affecting RuBP reactivity. Since these speculations are not well founded we refrain from expanding on them in the text.

I am curious as to how oxygenation kinetics could be captured in this, or a derivative screen.

We are excited to more carefully examine the effect of oxygenation in future studies and can share some preliminary data that provide additional context for the system employed here. Briefly, we have performed two exploratory titrations of O_2 , similar to the CO_2 titration reported in the present manuscript, one at 0.5% CO_2 and the other at 0.2% CO_2 . One striking result is that, just as increasing $[CO_2]$ led to a larger proportion of functional mutations, reducing the O_2 concentration resulted in a more dramatic improvement in the functional:non-functional mutant ratio. This result indicates that O_2 is acting as a burden for a significant proportion of mutations.

In the literature there exist a few examples of *R. rubrum* rubisco mutants with impaired specificity, including H44N and D117V (Mueller-Cajar et al. 2007). As one would hypothesize based on this impairment, the fitness of these mutations appears to decline with increasing $[O_2]$. We interpret this result to mean that oxygenation flux is deleterious to cell physiology and that mutants with increased propensity to oxygenate experience worse growth outcomes compared to WT as $[O_2]$ increases.

In the future, we hope that this type of data may allow us to predict all 4 kinetic parameters ($k_{cat,C}$, K_C , $k_{cat,O}$, and K_O) but will, of course, need to make many *in vitro* measurements to confirm such a procedure and calibrate it properly.

[REDACTED]

Referee #3

RuBisCO, one of the most abundant enzymes on Earth, plays a key role in global carbon fixation. Nevertheless, it is characterized by slow catalytic turnover and poor CO₂ affinity. Increasing its activity (and minimizing an undesired side reaction with molecular oxygen) could potentially improve photosynthetic efficiency and hence crop yields, but engineering RuBisCO has proved challenging. In this study, Prywes et al. have addressed this issue by applying genetic selection methods to map sequence-function relationships in a bacterial RuBisCO in high throughput. Using an engineered *E. coli* as a host, they successfully assayed >99% of all single amino acid substitutions in the enzyme, determining relative velocities and CO₂ affinity parameters for thousands of variants in parallel. While no enzymes with higher specific activities were found, a few substitutions were identified that conferred higher CO₂ affinity and maintained WT-like fitness levels. This system thus provides a powerful platform for high-throughput engineering and systematic exploration of the tradeoffs that constrain RuBisCO's evolution. The work appears to have been carried out with care and admirable attention to detail. If the concerns outlined below can be successfully addressed, the manuscript may therefore be suitable for publication in *Nature*.

Critique:

1) The acronyms Ri5P, Ru5P and RuBP in Fig.1 should be defined somewhere in the legend or the text.

We have updated the caption text on line 94.

2) Fig. 1 legend: k_{cat} is a rate constant, not a “catalytic rate”.

We have updated the caption text on line 99 to correct this error.

3) Fig.2 legend: panels C) and D) are mislabeled.

We have updated the caption to correct this error.

4) Lines 172-175: If no binding site exists for CO₂, how can one speak of “improved affinity”? Why would the kinetic plots show saturation if CO₂ does not bind to the enzyme? In such a case, wouldn't one expect a linear increase in rate with increasing CO₂ concentration?

The absence of a binding site for CO₂ is a peculiarity of rubisco that arises from the small and apolar nature of the CO₂ substrate. This feature of rubisco catalysis was established by an NMR study (Gutteridge et al. 1984) which showed that there is no CO₂-bound state of the enzyme. Subsequent structural studies have not identified a CO₂ binding site and the carboxylation of RuBP is thought to be irreversible (Lorimer et al. 1986; Tcherkez et al. 2018; Douglas-Gallardo et al.

2022). However, as noted, CO₂ and O₂-dependent enzyme kinetic analysis of rubisco exhibits classical substrate saturation and these two molecules can inhibit one another competitively.

The explanation for this seeming contradiction is explained by the rubisco mechanism which has multiple steps, 1) CO₂ addition is followed by 2) hydration and cleavage of the 6-carbon intermediate, 2-carboxy-3-keto-D-arabinitol 1, 5-bisphosphate (CKABP). As the CO₂ concentration increases, the catalytic rate becomes limited by the second step (Savir et al. 2010). In more typical enzymes the K_M value is generally thought to reflect the off-rate of bound, unreacted substrate. In rubisco, there is no pre-equilibrium of bound CO₂, and the apparent K_M value (K_C) is determined by the CO₂ on-rate. This on-rate constant, k_{on}, depends on the fraction of rubisco enzymes with activated RuBP (what fraction of the sugar is in the ene-diolate form), as well as other stereoelectronic factors affecting gas entry into the active site and subsequent bond formation. These factors can influence the accessibility of the bound RuBP and the energy of the ene-diolate.

If there were no second step in the mechanism, we concur with the Reviewer - the reaction velocity would increase linearly with increasing [CO₂] until some other step became limiting (i.e. enolization).

We have added one of the citations above and edited the sentence in the text to read:

Since the CO₂ addition step in catalysis is thought to be irreversible²⁹ and there is no binding site for CO₂ in the enzyme³⁰, this trend may be related to subtle changes in the electronics of the active site or the geometry of the bound sugar substrate before or during bond-formation with CO₂.

5) Fig. 3B: The population of active variants appears to shift to the left somewhat as the CO₂ concentration increases, suggesting that it becomes less fit. Is this expected? Is bacterial RuBisCO subject to autoinhibition under these conditions?

The population shifts to the right with increasing CO₂. We have updated the figure colors to make this trend more obvious. This effect is expected because a subpopulation of mutants have elevated K_C values which means that they become more fit as CO₂ levels rise to rescue them.

6) Fig. 3D and E: Does 1 mM RBP, the concentration used for the in vitro kinetic measurements, saturate RuBisCO? How does this concentration compare to the concentration of RBP in vivo? Could differences in the in vitro and in vivo RBP concentrations explain the “unexpectedly” lower K_C values measured in vitro (line 163)?

For *in vitro* measurements, 1 mM is saturating for the WT enzyme (K_{M,RuBP} = 15 μM in Gutteridge et al. 1984). We do not know the concentration of RuBP *in vivo*. If it were the primary limiting factor we would expect to see a plateau in the k_{cat} vs. fitness plot (Figure 1E, see response to Reviewer #1). If that were the case, any rubisco over a given k_{cat} threshold would be equally sufficient in the face of low PRK flux and RuBP limitation (Nguyen et al. 2024). We did not observe such a plateau, nor did we observe substantial growth rate differences when we induced PRK expression at different levels with anhydrotetracycline, but it is possible that RuBP levels are held constant through allosteric control (Sporre et al. 2022; Abdelal and Schlegel 1974).

The question of whether changes in K_{M,RuBP} could explain the unexpectedly lower K_C values is an interesting one. In order to answer it we measured the K_{M,RuBP} values for WT and our 2 selected mutants (Reviewer Figure 5). We found that K_{M,RuBP} for A102Y was lower than WT. It is possible

that that could explain the discrepancy in our predictions. However, V266T was found to have a $K_{M,RuBP}$ value that was very close to WT so the discrepancy in our prediction cannot be explained by RuBP binding differences in this case.

We include these data in a new Extended Data Fig. 9 and edit the manuscript (line 206) to reflect this observation along with additional methods (line 431).

Reviewer Figure 5: K_M determinations for key mutants. A-C) Results of RuBP titrations using the spectrophotometric rubisco activity assay. **D)** Calculated K_M values with Michaelis-Menten fitting errors indicated.

7) Line 190: why “also”? The three variants mentioned in the previous sentence show an apparent increase (better than “improvement”) in CO₂ affinity, not a decrease.

We apologize for the confusing wording and have removed “also” from the manuscript.

8) Line 199: “Our K_c predictions were isolated from expression effects because mutants were judged individually by their relative performance across a CO₂ titration and were thus more accurate.” This seems like an odd explanation. Isn’t K_c, by definition, independent of enzyme concentration and therefore expression effects?

That is the concept that we were trying to convey, we have edited the text to read:

Unlike V_{max} , the affinity parameter is independent of enzyme concentration so K_C predictions are expected to be more accurate.

9) Lines 200-203: Because k_{cat} for the characterized variants decreased more than CO₂ affinity increased, catalytic efficiency, defined as k_{cat}/K_m, dropped (Extended Data Table 2). This finding argues against the statement in the abstract that “non-trivial kinetic improvements are readily accessible.” The mutations do not constitute a kinetic improvement; as noted by the authors, high flux was likely maintained by increasing protein expression, which is much easier to accomplish than improving catalytic efficiency. Instead, the results appear to underscore the difficulty of engineering more efficient RuBisCOs, at least through single amino acid substitutions, and speak to the need to screen more complex libraries containing combinations of mutations to find variants with improved catalytic efficiency.

We apologize for the implication that we found mutations that improved catalytic efficiency, as noted, none of the mutants characterized in this study had improved $k_{cat,C}/K_C$. What our results do show is that carboxylation affinity can be substantially increased with point mutations. Indeed it will only be possible to discover mutants with improved catalytic efficiency by screening larger

libraries, if at all. We have edited the abstract (line 34), introduction (line 66) and conclusion (line to refer to these as non-trivial changes or alterations in biochemistry. We also note that libraries of more complex mutants may reveal mutants with improved efficiencies.

10) How do the mutations that increase CO₂ efficiency affect the competing oxygenation reaction? Given the importance of suppressing this undesired side reaction, the authors should measure the CO₂ vs O₂ specificity.

We thank the reviewers for suggesting this important further inquiry. Since specificity is determined as a combination of four parameters it could change in either direction as a result of the changes to $k_{cat,C}$ and K_C measured in this study. The preliminary data above indicated that specificity was likely constant because the fitness of these mutants did not vary substantially as a function of O₂ concentration. To confirm this, we measured the specificity of each of these mutants along with appropriate controls from the literature using Membrane Inlet Mass Spectrometry (MIMS). In a new Extended Data Figure 9 we present these results which show that these mutations did not have a significant effect on specificity. We conclude that the ratio between $k_{cat,O}$ and K_O must shift to match the new ratio between $k_{cat,C}$ and K_C such that $S_{C/O}$ is similar to that of the WT enzyme in both cases. This seeming coincidence is in agreement with the observation that $S_{C/O}$ does not vary considerably between rubiscos from different species (Flamholz et al. 2019).

We have edited the text to include this new result (lines 236-239) and added a methods section for the MIMS (lines 461-479).

Reviewer Figure 6: Specificity values measured by membrane inlet mass spectrometry in triplicate. Comparisons to literature values are displayed when available. Literature data for WT is from (Iñiguez et al. 2021). Error bars represent the SEM of all measurements compiled in that published analysis. Literature data for H44N and D117V is from (Mueller-Cajar et al. 2007). Error is taken from table 2 in that publication. * indicates $p < 0.05$ by a Welch's t-test in comparison to WT

Updated biochemistry table (now in Extended Figure S9)

	k_{cat}	k_{cat} error	K_C	K_C error	$S_{C/O}$	$S_{C/O}$ error	$K_{M,RuBP}$	$K_{M,RuBP}$ error
WT	9.9	0.4	149	10	11	1	16.6	0.7
A102Y	1.71	0.04	53	3	9	2	8	2
V266T	5.1	0.1	87	5	10.8	0.6	16	1

Dear Editor

Please find our resubmitted manuscript, which we hope addresses all of the remaining technical concerns and specifically addresses Nature's data and editorial checklist items. These are all enumerated with a brief authors' explication in the below postscript. Our response is in green in this point by point document whereas the changes (in track changes mode) are in blue in the manuscript document.

Briefly, I would also like to bring your attention to two details. First, we have attempted to ameliorate the remaining technical issue, which relates to Reviewer 3's concern regarding the use of the term affinity. Our response, fully delineated below, attempts to harmonize the rubisco community's treatment of the issue in the context of enzymology broadly. We believe our compromise on terminology is sufficient. Secondly, given the overall helpful feedback and this item in particular, we do think it would be useful to have the reviews and responses public and are happy to participate in the transparent review process.

If we have missed any remaining minor issues, please let us know and we will address as quickly as possible.

Regards,

Dave Savage

1. Please provide a version of the text with changes tracked so we can assess whether the point made by referee 3 has been appropriately addressed.

Text changed from the original submission to the resubmission is left in blue. Text changed for this second resubmission is indicated in green in the word doc in addition to changes being tracked.

Reviewer 3 concern:

The one remaining issue regards the purported absence of a CO₂ binding site in RuBisCo. If true, equating the K_c parameter with CO₂ "affinity", as is done consistently throughout the manuscript, should be avoided. This is an unnecessary and misleading simplification. As clearly explained in the rebuttal, there is no pre-equilibrium of bound CO₂, so K_c -- formally the CO₂ concentration at which the reaction velocity is half maximal -- is not the dissociation constant for CO₂. Instead, this parameter is determined by the CO₂ on-rate, the subsequent rate of its reaction with activated RuBP, and presumably the binding constant for RuBP.

Reviewer 3 is pointing out that rubisco catalysis has an unusual feature where one of the substrates is never non-covalently bound by the enzyme. This is a feature of many (but not all) gas-dependent enzymes including Cytochrome P450 (Denisov et al. 2005), Hydrogenase, CO dehydrogenase, Nitrogenase and others (Fontecilla-Camps et al.). It is typical in these cases for the word affinity to be used (though in some of these cases the enzyme-binding may still be reversible).

This line of concern reflects a debate in the literature in the early 20th c. regarding the derivation of the Michaelis-Menten equation. In the original derivation it was assumed that the substrate off-rate was much faster than the forward rate of catalysis ($k_{off} \gg k_{cat}$). An alternative derivation was later found that demonstrated that that assumption is unnecessary and k_{off} can be, as it is in the case of rubisco, effectively 0 (Briggs and Haldane 1925). As the reviewer points out, K_M is then dependent on factors other than k_{off} , most significantly k_{on} .

Regarding the use of the word affinity, it is unclear what word would be more appropriate. We seek to remain consistent with four decades of consistent use of the word in the rubisco literature since the discovery of the absence of a Michaelis-complex. The following papers are a small selection across 40 years from a variety of key authors in the field, they all use the term “affinity, including in titles and abstracts, mostly without qualification: Jordan and Ogren 1984, Pierce et al. 1986, Badger et al. 1998, Tcherkez et al. 2006, Whitney et al. 2009, Tcherkez et al. 2016. While some authors occasionally qualify the term as “apparent” affinity, that is frequently a reflection of the phenomenological nature of the measurement. We have added that qualification to the abstract and at line 158 to further emphasize the unusual nature of the concept.

Denisov IG, Makris TM, Sligar SG, Schlichting I. Structure and chemistry of cytochrome P450. *Chem Rev.* 2005;105: 2253–2277.

Fontecilla-Camps JC, Amara P, Cavazza C, Nicolet Y, Volbeda A. Structure-function relationships of anaerobic gas-processing metalloenzymes. *Nature.* 2009;460: 814–822.

Briggs GE, Haldane JB. A note on the kinetics of enzyme action. *Biochem J.* 1925;19: 338–339.

Jordan DB, Ogren WL. The CO₂/O₂ specificity of ribulose 1,5-bisphosphate carboxylase/oxygenase. *Planta.* 1984;161: 308–313.

Pierce J, Lorimer GH, Reddy GS. Kinetic mechanism of ribulosebisphosphate carboxylase: evidence for an ordered, sequential reaction. *Biochemistry.* 1986;25: 1636–1644.

Badger MR, John Andrews T, Whitney SM, Ludwig M, Yellowlees DC, Leggat W, et al. The diversity and coevolution of Rubisco, plastids, pyrenoids, and chloroplast-based CO₂-concentrating mechanisms in algae. *Canadian Journal of Botany.* 1998. pp. 1052–1071. doi:10.1139/b98-074

Tcherkez GGB, Farquhar GD, Andrews TJ. Despite slow catalysis and confused substrate specificity, all ribulose bisphosphate carboxylases may be nearly perfectly optimized. *Proc Natl Acad Sci U S A.* 2006;103: 7246–7251.

Whitney SM, Kane HJ, Houtz RL, Sharwood RE. Rubisco oligomers composed of linked small and large subunits assemble in tobacco plastids and have higher affinities for CO₂ and O₂. *Plant Physiol.* 2009;149: 1887–1895.

Tcherkez G. The mechanism of Rubisco-catalysed oxygenation. *Plant Cell Environ.* 2016;39: 983–997.